# From typhoon rainfall to slope failure: optimizing susceptibility models and dynamic thresholds for landslide warnings in Zixing City, China

**Weifeng Xiao[1], Guangchong Yao[1], Zhenghui Xiao[1], Ge Liu[2*], Luguang Luo[1], Yunjiang Cao[1], Wei Yin[3]**

[1]School of Earth Sciences and Spatial Information Engineering, Hunan University of Science and Technology, Xiangtan 411201, China

[2]Northeast Institute of Geography and Agroecology, CAS, Changchun 130102, China

[3]Hunan Institute of Geological Disaster Investigation and Monitoring, Changsha 410004, China

Corresponding author.

E-mail address: liuge@iga.ac.cn (Ge Liu)

**Abstract**: Typhoon-specific rainfall-induced landslides pose critical hazards in mountainous regions, yet existing warning systems inadequately capture the distinct rainfall dynamics of these extreme events. To address this limitation, we propose an integrated framework combining optimized susceptibility predictions with dynamic rainfall thresholds tailored to typhoon patterns. The approach enhances machine learning accuracy through buffer-based negative sampling and variable weighting. It also introduces a spatiotemporal rainfall analysis to distinguish between short-term intense downpours and cumulative soil saturation. Tested in Zixing City, Hunan Province, China, where over 700 landslides were triggered by Typhoon Gaemi, the framework proved effective. The support vector machine (SVM) model achieved the best performance using frequency ratio (FR) inputs with a 0.5 km buffer (F1-score: 0.859, AUC: 0.914), correctly classifying 86.4% of landslides as high or very high susceptibility. The rainfall analysis identified 24-hour intensity combined with 7-day antecedent rainfall as the optimal trigger, effectively capturing both immediate and cumulative moisture effects. Spatially, rhyolite and granite slopes and areas near roads emerged as hotspots for failure

(distance < 800 m, FR = 1.499 for roads; FR = 1.546 for rhyolite). The integrated warning
system shows high spatial efficiency, with high-risk areas covering only 34.2% of the study
region yet capturing 71.4% of historical landslides. Additionally, the framework generated
high-risk zone maps that align strongly with historical events. This work highlights the unique
nature of typhoon-driven slope instability and provides a transferable framework for disaster
risk reduction in cyclone-prone regions.
**Keywords:** Typhoon-induced landslide; Slope failure; Hazard warning system; Dynamic
thresholds; Landslide susceptibility mapping
**1 Introduction**
Landslides pose significant threats to mountainous regions globally (Froude and Petley,
2018), especially in areas where steep terrain, complex geology (Thiene et al., 2017), and
extreme weather events like typhoons intersect. In Southeast China, typhoon-induced
landslides have become a growing concern due to the region's rapid urbanization and the
increasing variability in climate patterns (Gariano and Guzzetti, 2016; Fan et al., 2018). The
Nanling Mountains, in southern China, are particularly vulnerable to landslides due to a
combination of extreme topographic relief and complex geological conditions during the
typhoon season (Zou et al., 2023).
Typhoons typically bring prolonged antecedent rainfall, followed by intense, short bursts
of precipitation (Li et al., 2019). These conditions create unique hydrological environments
that exceed the complexity of typical rainfall-triggered landslides (Chung and Li, 2022).
These events trigger slope failures through cumulative soil saturation and sudden hydrological
stress, challenging traditional landslide prediction methods (Yang et al., 2017). Despite
advances in landslide susceptibility prediction (LSP) and rainfall threshold modeling, current
approaches remain inadequate. Three critical limitations persist: severe data imbalance effects,
suboptimal integration of variable selection with machine learning algorithms, and lack of
spatially-explicit rainfall thresholds for typhoon-specific conditions (Segoni et al., 2018a;
Regmi et al., 2024).

Most existing studies employ ad-hoc buffer distances without systematic optimization,

leading to inconsistent model performance across different geological settings (Lombardo and
Mai, 2018). Traditional methods attempt to mitigate this imbalance by randomly sampling
non-landslide points across the study area (Steger et al., 2016; Dou et al., 2023). However,
random selection can introduce spatial bias, as non-landslide points might include areas that
are unstable but have not yet been identified as landslide-prone (Kalantar et al., 2018).

To address this limitation, more recent approaches have employed buffer-based negative

sampling, which systematically excludes non-landslide points near known landslide sites.
This method assumes that adjacent areas share similar environmental conditions (e.g., slope,
lithology) and therefore should not be classified as "stable" (Achu et al., 2022). Several
studies have tested varying buffer distances, ranging from tens to thousands of meters, to
determine the optimal distance for different regions. However, systematic evaluation of buffer
distance optimization coupled with variable weighting methods remains largely unexplored.

LSP is primarily focused on identifying areas prone to slope failure, based on static

environmental factors such as topography, lithology, land cover, and hydrology (Zêzere et al.,
2017; Guo et al., 2024). Traditional approaches to LSP often rely on deterministic and
statistical methods, including information value (IV), certainty factor (CF), frequency ratio
(FR), logistic regression (LR), and weight of evidence (WOE). These methods quantify the
relationship between historical landslide occurrences and predisposing factors using linear or
semi-linear approaches (Ciurleo et al., 2017; Reichenbach et al., 2018). However, these
methods oversimplify the complex, nonlinear interactions that govern slope stability
(Merghadi et al., 2020).
In contrast, machine learning (ML) algorithms, such as support vector machine (SVM)
and light gradient boosting machine (LightGBM), have emerged as powerful alternatives.
SVM excels in high-dimensional classification tasks and effectively identifies optimal
hyperplanes separating landslide-prone from stable areas (San, 2014; Huang and Zhao, 2018).
LightGBM offers superior scalability and computational efficiency for processing large
geospatial datasets (Sun et al., 2023). Both SVM and LightGBM capture intricate
relationships among variables without restrictive assumptions, making them superior to
traditional methods in terms of predictive accuracy (Yang et al., 2023). However, frameworks
that systematically integrate variable weighting methods with advanced ML algorithms for
LSP optimization are lacking.
For temporal prediction, existing rainfall threshold approaches predominantly use
generalized regional thresholds that inadequately capture local geological heterogeneity and
typhoon-specific rainfall patterns (Guzzetti, 2021; Banfi and De Michele, 2024). These
thresholds are typically defined based on cumulative or intensity-duration (I-D) rainfall values
(Piciullo et al., 2017; Segoni et al., 2018a). In typhoon-prone regions, dynamic rainfall
thresholds are crucial due to the unique combination of long-duration antecedent rainfall and
sudden high-intensity bursts of precipitation (Guzzetti et al., 2020). Traditional empirical
methods fail to provide spatially continuous threshold surfaces that account for local
environmental variability (Piciullo et al., 2018).
Recent advances have integrated multi-temporal rainfall parameters with advanced
statistical techniques to optimize rainfall thresholds (Segoni et al., 2015; Huang et al., 2022),
accounting for diverse triggering mechanisms. Additionally, spatial interpolation methods,
such as Kriging, have been applied to generate continuous rainfall threshold surfaces that
allow for local variations in geological and environmental conditions (Kenanoglu et al., 2019;
Segoni et al., 2018b). This approach, when combined with high-resolution susceptibility maps,
contributes to the development of integrated hazard warning systems that can dynamically
adjust to typhoon-specific rainfall-induced scenarios (Piciullo et al., 2018; Mirus et al., 2018).
This study examines Zixing City, a mountainous region in southeastern Hunan Province,
frequently affected by typhoon-induced extreme rainfall. Its steep slopes, fractured geology,
and high sensitivity to rapid pore-pressure increase render it particularly vulnerable (Ma et al.,
2025). The large number of landslides (>700) triggered by Typhoon Gaemi in July 2024
provides a valuable dataset for model calibration and validation.
Here we developed an integrated framework that combines (i) optimized buffer distances
for negative sampling, (ii) bivariate weighting methods (IV, CF, FR) with advanced machine
learning classifiers (SVM, LightGBM), and (iii) spatially continuous, typhoon-specific
rainfall thresholds derived through Kriging interpolation. The specific objectives are to (1)
determine optimal buffer distances that minimize spatial bias in imbalanced datasets, (2)
evaluate the performance gain from coupling bivariate weights with machine learning
algorithms, (3) establish dynamic rainfall thresholds suited to typhoon rainfall patterns, (4)
generate continuous threshold surfaces via Kriging, and (5) integrate high-resolution
susceptibility maps with these thresholds to support an operational early warning system. This
approach improves landslide prediction in typhoon-prone mountainous regions and provides a
transferable methodology for similar environments.
**2   Study area and data sources**
**2.1   Study area**
Zixing City (25°34′–26°18′ N, 113°08′–113°44′ E), covering 2,747 km² in southeastern
Hunan Province, China (Fig. 1), is located within the Nanling Mountains geological province.
Situated approximately 400 km inland from the South China Sea, Zixing lies at the
intersection of the Nanling Mountains and low hills, forming a watershed divide between the
Yangtze and Pearl River basins. The region is characterized by steep topography, with
elevations ranging from 125 to 1,691 meters and slopes exceeding 30° across 78% of the area.
This mountainous terrain, combined with fractured geology and active NE-SW trending faults
such as the Chaling-Yongxing Fault Zone, creates a permeable fracture network that
facilitates groundwater drainage.
The climate of Zixing is subtropical monsoon, with annual precipitation averaging 1,550
mm, 70% of which occurs from April to September. Typhoons significantly contribute to
rainfall, inducing rapid pore-pressure increases in shallow aquifers (3–8 m depth). These
climatic and geological conditions make Zixing particularly vulnerable to landslides,
providing a valuable context for this study. The extensive landslide dataset triggered by
Typhoon Gaemi in July 2024 (>700 events) serves as a critical resource for model calibration
and validation.

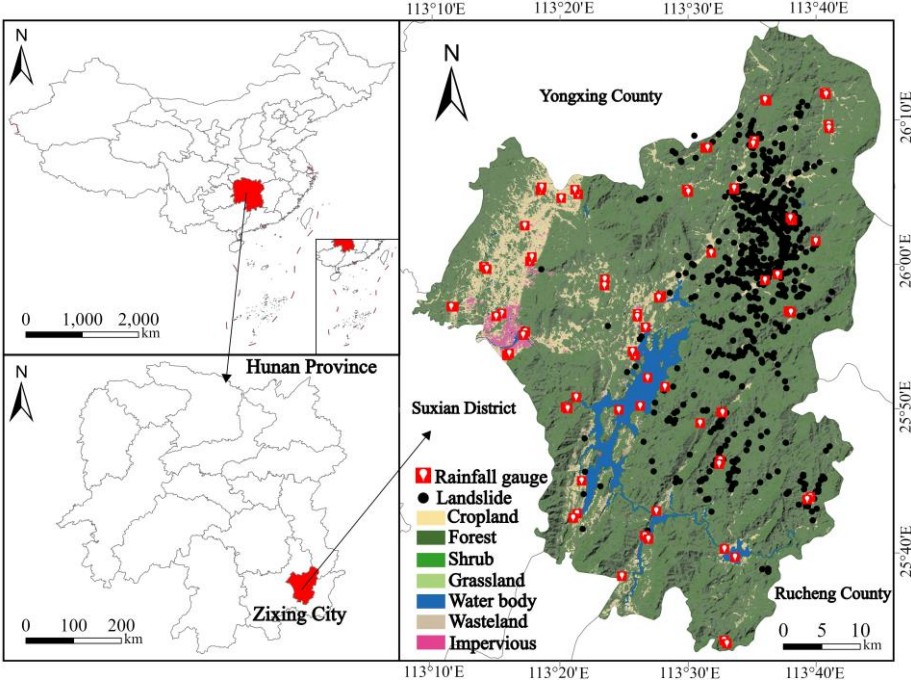

**Figure 1** Geographical distribution of the study area, landslides and rainfall gauges.
**2.2 Data collection and preprocessing**
**2.2.1 Compilation of landslide catalogue**
A comprehensive inventory of 705 landslide events triggered by Typhoon Gaemi on July
27, 2024, was compiled from the Hunan Center for Natural Resources Affairs. The landslide
locations were verified through field inspections and high-resolution satellite imagery to
ensure spatial accuracy and completeness of the dataset.

### 2.2.2 Landslide conditioning factors and data sources

Based on extensive literature reviews and the geoenvironmental characteristics of the
study area, twelve conditioning factors were selected for landslide susceptibility analysis:
elevation, slope gradient, slope orientation, curvature, topographic wetness index (TWI),
stream power index (SPI), normalized difference vegetation index (NDVI), distances to roads,
rivers, and faults, and lithology (Fig. 2).
Topographic factors (elevation, slope gradient, slope orientation, TWI, SPI, and
curvature) were extracted from a 30-meter digital elevation model (DEM) obtained from the
Geospatial Data Cloud (https://www.gscloud.cn). Environmental factors including NDVI and
proximity variables (distances to roads, rivers, and fault lines) were derived from 1:50,000-
scale cartographic maps and Landsat 8 OLI imagery from the same platform. Geological
composition and structural data were acquired from 1:100,000-scale geological maps.
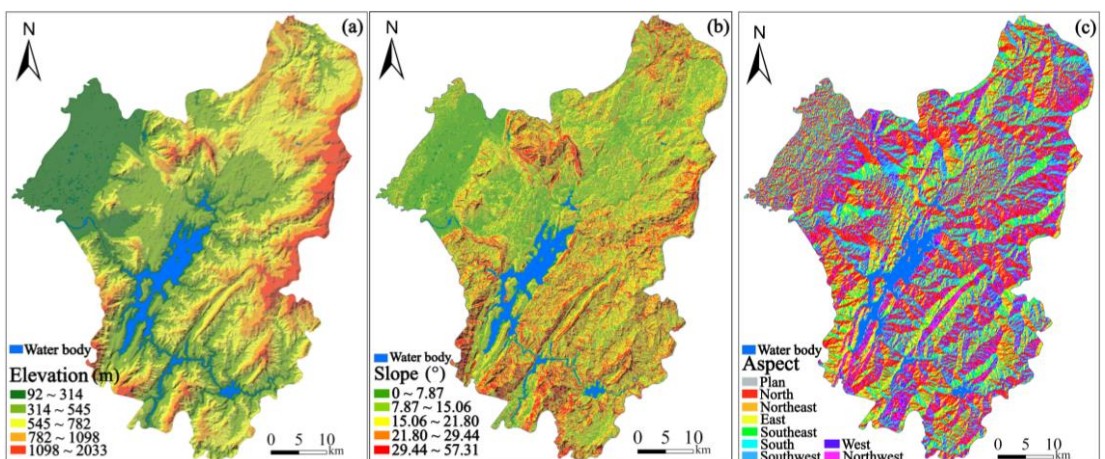


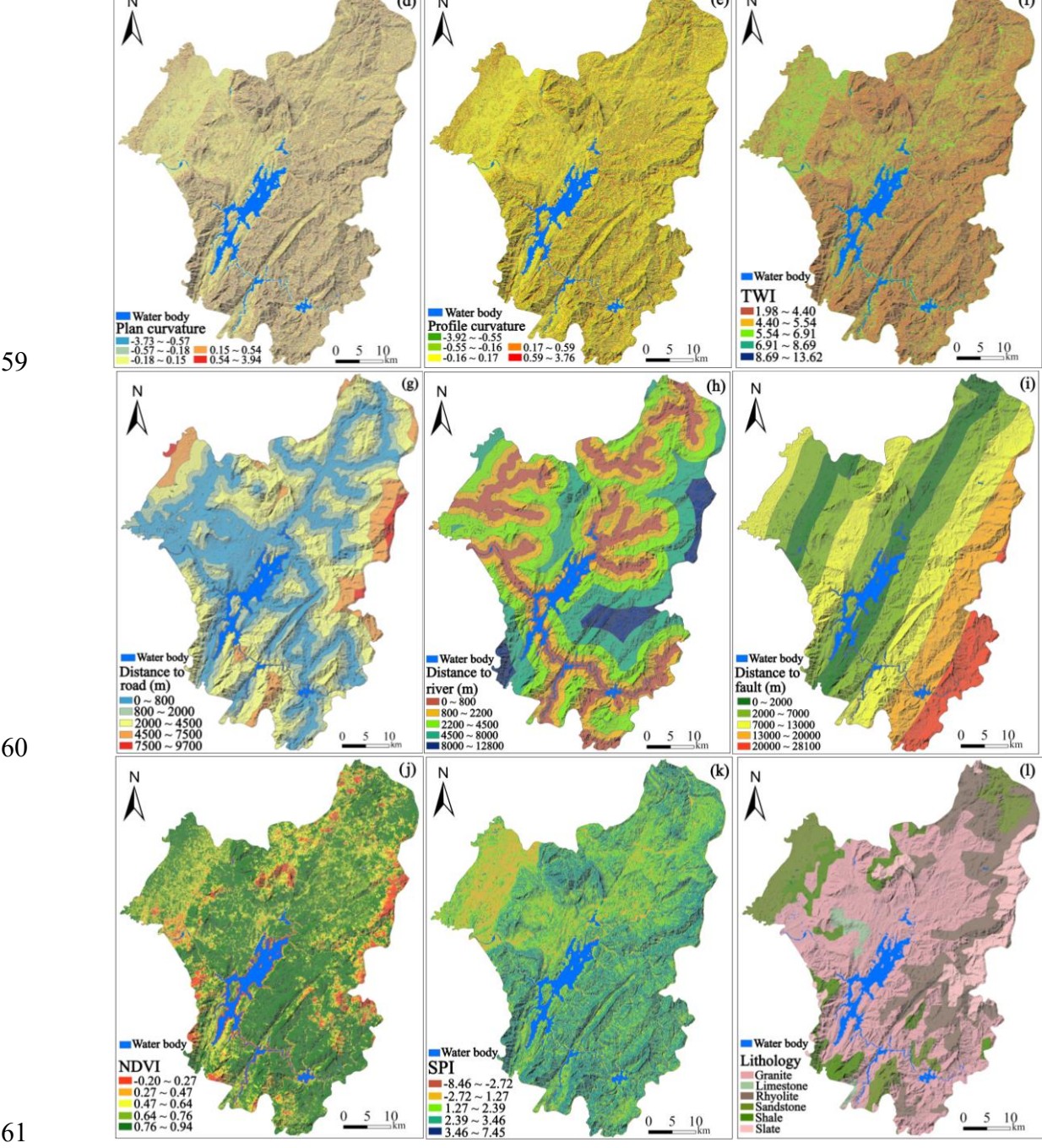


**Figure 2** Landslide-related conditioning factors.

**2.2.3 Data preprocessing and spatial standardization**
We transformed all conditioning factors into continuous statistical measures using IV,
CF, and FR methods and then resampled them to a uniform 60-meter resolution. This
resolution was selected to balance computational efficiency with scale appropriateness for
regional landslide analysis while maintaining compatibility with the available geological map
scale (1:100,000).

The study area was divided into $60 \times 60$ meter grid cells, with landslides smaller than the

grid resolution aggregated to the nearest cell centroid. Multiple landslides within a single cell

were treated as one event to maintain spatial independence required for machine learning

modeling. This preprocessing approach ensures statistical validity by minimizing spatial

autocorrelation effects while providing adequate representation of landslide distribution

patterns across the study area.

**2.2.4 Rainfall data collection and spatial distribution**

Rainfall data for the study were obtained from 12 automatic rain gauge stations

strategically distributed across Zixing City and its surrounding areas (Fig. 1). These stations,

operated by the Hunan Meteorological Administration, provided hourly precipitation records

during Typhoon Gaemi (July 20–30, 2024) and the preceding antecedent period. The spatial

distribution of gauge stations ensured adequate coverage of the study area's topographic and

climatic gradients.

To assign rainfall parameters (H1, H12, H24, H72, and D7) to each of the 705 landslide

points, we employed the Kriging interpolation to generate spatially continuous rainfall

surfaces from discrete gauge measurements. This geostatistical method accounts for spatial

autocorrelation in rainfall patterns and provides optimal unbiased estimates by weighting

nearby observations based on their spatial proximity and correlation structure.

Spherical variogram models were fitted to the rainfall data through iterative optimization,

with model selection based on minimum Akaike Information Criterion (AIC) values. The

interpolation accuracy was rigorously evaluated through leave-one-out cross-validation,

where each gauge station was sequentially removed and its rainfall values predicted using the

remaining 11 stations. Four statistical metrics were used to assess performance: Root Mean

Square Error (RMSE), Mean Absolute Error (MAE), correlation coefficient (R), and Nash-

Sutcliffe Efficiency (NSE).


**Table 1** Kriging interpolation accuracy assessment for rainfall parameters.

| Parameter | RMSE (mm) | MAE | R | NSE |
|-----------|-----------|------|------|------|
| H1 | 4.2 | 3.1 | 0.76 | 0.71 |
| H12 | 11.7 | 8.9 | 0.83 | 0.78 |
| H24 | 16.3 | 12.6 | 0.87 | 0.82 |
| H72 | 24.8 | 18.4 | 0.81 | 0.77 |
| D7 | 29.6 | 22.7 | 0.78 | 0.73 |

The validation results demonstrated acceptable interpolation accuracy across all rainfall
parameters, with correlation coefficients ranging from 0.76 to 0.87 and Nash-Sutcliffe
Efficiency values between 0.71–0.82. Despite some limitations inherent to the sparse gauge
network in mountainous terrain, the interpolation performance was deemed sufficient for
regional landslide susceptibility analysis, ensuring reasonable spatial representation of
precipitation patterns across the study area.
**3  Methodologies**
This study proposes an integrated framework for optimizing LSP and typhoon-specific
rainfall thresholds within hazard warning systems (Fig. 3). The framework includes the
following key components: (1) landslide susceptibility prediction and mapping, utilizing
twelve conditioning factors prioritizing typhoon-induced hydrological responses (e.g., TWI,
SPI) and 705 landslide records from July 27, 2024, optimized with five buffer distances and
evaluated using ROC curves; (2) dynamic rainfall threshold modeling based on typhoon
rainfall parameterization, validated and spatially interpolated using Kriging; and (3) the
integration of spatial and temporal probabilities to develop a typhoon-specific rainfall-induced
landslide warning system, demonstrated through a case study in Zixing City.

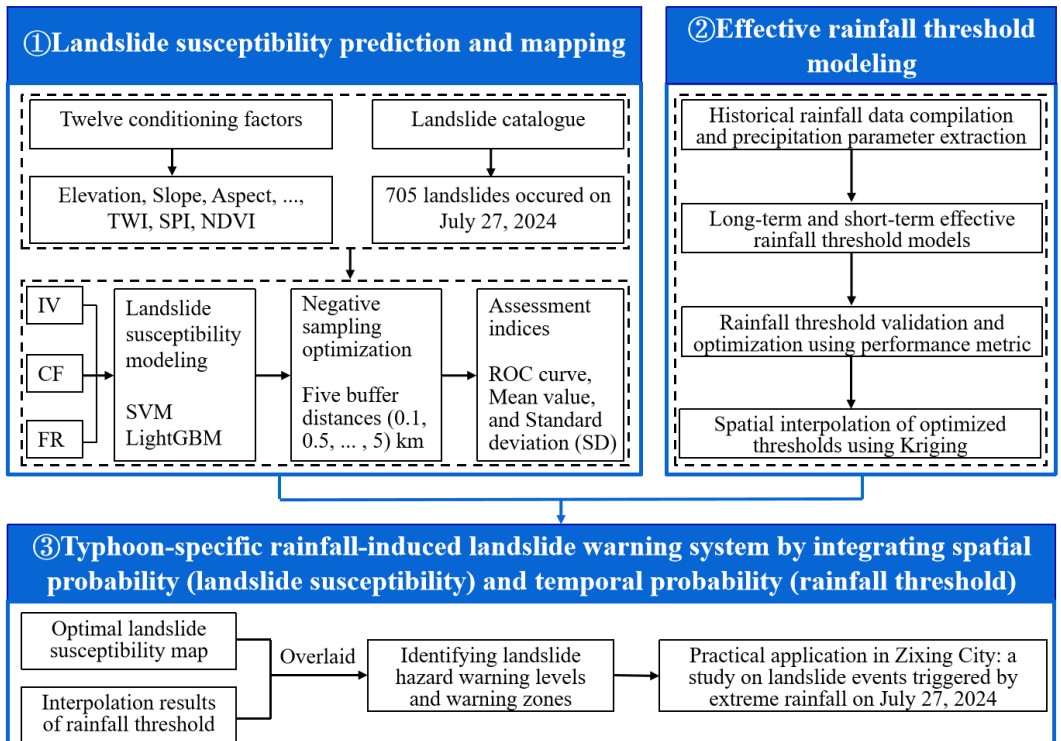

**Figure 3** Technical framework for developing a typhoon-specific rainfall-induced landslide warning system.

### 3.1 Landslide susceptibility prediction and mapping

### 3.1.1 Machine learning models: selection rationale and implementation

We selected SVM and LightGBM to address three key challenges in typhoon-specific rainfall-induced landslide prediction: (1) severe class imbalance (landslides <0.5% of study area), (2) complex non-linear interactions between rainfall and terrain factors, and (3) computational efficiency for operational early warning.

SVM excels in binary classification with limited samples through structural risk minimization (Kalantar et al., 2018; Wang et al., 2020), making it suitable for typhoon-triggered landslide mapping. Its margin-maximization approach handles the class imbalance between stable and landslide areas, while the RBF kernel captures localized failure patterns under concentrated typhoon rainfall. The regularization parameter $C$ prevents overfitting to specific typhoon events, ensuring model transferability. The SVM optimization problem is defined as:

$$\min_{w,b,\xi} \frac{1}{2} w^T w + C \sum_{i=1}^{n} \xi_i \qquad (1)$$

subject to the constraint:
$$y_i(w^T \phi(x_i) + b) \geq 1 - \xi_i, \quad \xi_i \geq 0, \quad i = 1, \cdots, n \qquad (2)$$

where $w$ is the normal vector to the hyperplane, $b$ is the bias term, $\xi_i$ are slack variables,
$\phi(x_i)$ maps input vectors to a higher-dimensional space, and $y_i$ denotes the class label (-1 or 1)
for each sample $x_i$. We optimized the RBF kernel parameters using grid-search with 5-fold
cross-validation, where $C \in [0.1, 100]$ and $\gamma \in [0.001, 1]$. Across all configurations (three
input methods × five buffer distances), optimal values varied as follows: $C = 5$–$15$ and $\gamma =$
$0.10$–$0.25$, with median values of $C = 10$ and $\gamma = 0.15$.

LightGBM complements SVM through gradient boosting with sequential error

correction, offering distinct advantages for regional-scale landslide mapping. Its histogram-
based algorithm enables efficient processing of large spatial datasets (Sun et al., 2023; Sahin,
2020). Additionally, LightGBM automatically captures complex feature interactions. The
minimized objective function is expressed as:
$$L = \sum_{i=1}^{N} (y_i - \hat{y}_i)^2 + \lambda \sum_{j=1}^{M} \left\| \theta_j \right\|^2 \qquad (3)$$

where $y_i$ is the true label, $\hat{y}_i$ is the predictive value, $\lambda$ is a regularization parameter, and $\theta_j$
represents the parameters of the model. We optimized LightGBM hyperparameters through
Bayesian optimization. The optimal hyperparameters ranged as: num_leaves = 25–35,
learning_rate = 0.03–0.08, and max_depth = 6–10. Early stopping with a 50-round patience
window resulted in model convergence at 120–220 trees across different scenarios.
**3.1.2   Input variable weighting methods**

The IV method, grounded in information theory, assesses how different factors

contribute to landslide susceptibility within a study area (Niu et al., 2024). Factors such as

distance to roads and lithology were weighted higher in Zixing City due to their interaction with typhoon-induced soil saturation. The IV for each evaluation factor is determined using the formula below:

$$IV(F_i, K) = \ln \frac{N_i / N}{S_i / S} \tag{4}$$

where $IV(F_i, K)$ is the information value of evaluation factor $F_i$ in relation to landslide event $K$, $N_i$ refers to the number of landslides, $N$ is the total number of landslides, $S_i$ represents the area covered by factor $F_i$, and $S$ is the total area of the study area.

The CF method is a widely utilized probabilistic technique for assessing the likelihood of landslide occurrences (Zhao et al., 2021). It quantifies the prior probability of a landslide initiation under specific conditions of influential factors, utilizing spatial data from known landslide locations. The expression of CF is as follows:

$$CF = \begin{cases} \dfrac{PP_a - PP_s}{PP_s(1 - PP_a)}, & PP_a < PP_s \\ \dfrac{PP_a - PP_s}{PP_a(1 - PP_s)}, & PP_a \geq PP_s \end{cases} \tag{5}$$

where $CF$ is the certainty factor indicating the degree of association between an influential factor and potential landslide occurrence. It is derived from two area-proportional measures: $PP_a$, the proportion of landslide points within a specific factor class (number of landslide points in the class / total area of the class); and $PP_s$, the proportion of landslide points across the entire study region (total number of landslide points / total area of the region).

The FR is a prevalent method in statistical analysis that assesses the relative impact of various factors on the incidence of landslides (Panchal et al., 2021). An elevated FR value denotes a more significant influence of a factor on the likelihood of landslides. The FR is determined by the following equation:

$$FR = \frac{N_i / N}{S_i / S} \tag{6}$$

where *FR* is the frequency ratio, $N_i$ represents the number of landslides within the area
corresponding to the conditioning factor, $N$ is the total number of landslides, $S_i$ is the area
covered by the conditioning factor and $S$ is the total area of the study region.
**3.1.3 Buffer distance optimization and uncertainty assessment for LSP**
To generate negative (non-landslide) samples for LSP, areas within buffer distances of d
= 0.1, 0.5, 1.0, 2.0, and 5.0 km around landslide locations were excluded, with balanced
negative samples (n = 705) randomly selected from remaining stable areas for each distance.
The optimal buffer distance was determined by evaluating SVM and LightGBM model
performance using AUC, Precision, Recall, and F1-score metrics.
The selection of buffer distances (0.1–5.0 km) was based on Zixing's geomorphological
considerations and practices commonly reported in LSP. This range encompasses multiple
spatial scales: slope-scale processes (0.1–0.5 km), catchment-scale features (1.0–2.0 km), and
regional-scale geological units (5.0 km). The evaluation ensures optimal spatial representation
without a priori assumptions about scale dependencies (Chang et al., 2023).
Prediction uncertainty was assessed using the mean and standard deviation (SD) of
predicted landslide susceptibility values. Lower mean and SD values indicate reduced
prediction uncertainty and more concentrated susceptibility patterns, suggesting higher model
confidence in LSP (Huang et al., 2022), thereby complementing the buffer distance
optimization process.
**3.2  Effective rainfall threshold modeling**
**3.2.1  Rainfall parameterization and threshold calculation**
Typhoon-induced landslides are generally influenced by a combination of antecedent
moisture conditions and immediate precipitation, rather than by isolated rainfall events
(Mondini et al., 2023; Tufano et al., 2021). To account for the cumulative impact of multi-day

rainfall while incorporating hydrological processes such as evapotranspiration and drainage, we adopted the concept of effective rainfall ($P_e$), calculated as:

$$P_e = \sum_{i=0}^{n} k^i P_i \qquad (7)$$

where $P_i$ represents the daily rainfall on the $i$-th day preceding landslide occurrence, $n$ denotes the number of antecedent days considered, and $k$ is the effective rainfall decay coefficient (Segoni et al., 2018a). For hourly rainfall parameterization, $P_i$ is derived as:

$$P_i = \sum_{j=1}^{24} R_{ij} \qquad (8)$$

where $R_{ij}$ is the hourly rainfall at the $j$-th hour of the $i$-th day.

### 3.2.2 Long-term and short-term rainfall parameters

Rainfall-triggered landslides are generally triggered by two dominant mechanisms: prolonged low-intensity rainfall and short-duration high-intensity storms. Based on statistical analysis of historical landslide events in Hunan Province (Xiao et al., 2025), a 7-day antecedent period was identified as optimal for characterizing long-term rainfall impacts. Consequently, the 7-day effective rainfall (D7) was selected as the long-term parameter. Short-term rainfall metrics were defined as cumulative precipitation over 1 hour (H1), 12 hours (H12), 24 hours (H24), and 72 hours (H72) preceding landslide initiation. These intervals capture distinct rainfall characteristics: H1 reflects extreme short-term intensity for rapid slope failures, H12 and H24 represent sub-daily to daily precipitation critical for intermediate responses, and H72 accounts for multi-day storm sequences.

### 3.2.3   Rainfall threshold model development

The threshold modeling framework comprises three sequential steps:

(1) Parameter calculation: For each landslide sample, short-term rainfall parameters (H1, H12, H24, and H72) and the long-term rainfall parameter (D7) are calculated. The ratios of

short-term parameters to the long-term parameter are computed as: R1=H1/D7, R12=H12/D7,
R24=H24/D7, and R72=H72/D7.
(2) Threshold setting: Long-to-short-term ratio coefficients (RC1, RC12, RC24, and
RC72) are introduced as thresholds to determine the dominant rainfall pattern for each
landslide. These thresholds are used to classify landslides into short-term or long-term
Typhoon-induced categories.
(3) Coefficient optimization: A cyclic trial-and-error method is employed to determine
the optimal ratio coefficients (RC1, RC12, RC24, and RC72), maximizing the accuracy and
reliability of the model.
**3.2.4 Optimal ratio coefficient threshold determination**
The process of determining the optimal long-to-short-term ratio coefficient threshold is
demonstrated using H12-D7 as an example. The process for the remaining coefficients (H1-
D7, H24-D7, and H72-D7) follows a similar approach. A 5-fold cross-validation method is
applied, with the following procedure:
(1) Rainfall data extraction for landslide locations: For each of the 705 landslide points,
R12 and D7 values are extracted from these interpolated surfaces at the exact landslide
coordinates, ensuring that each landslide location receives rainfall values derived from the
spatially weighted contributions of all nearby gauge stations. R12 and D7 values for each
landslide are calculated using Equations (7) and (8).
(2) Data preparation: The dataset is divided into five equal parts for cross-validation,
with each part serving as a test set while the remaining four serve as the training set.
(3) Initial threshold setting: An initial threshold for RC12 is set based on the minimum
value in the training set.
(4) Threshold evaluation: For each fold, the RC12 threshold is compared with the R12
value of samples in the test set. If RC12<R12, the prediction is considered a failure.
Prediction accuracy is calculated for each RC12 threshold, adjusting in 0.001 increments until
the highest prediction accuracy is achieved.
(5) Optimal RC12 threshold determination: The RC12 threshold with the highest
prediction accuracy is selected for each fold. The final RC12 threshold is determined by
averaging the optimal thresholds from all five folds.
### 3.2.5   Spatial distribution of optimal threshold
According to the optimal ratio coefficient threshold determined in section 3.2.4 and the
long-term and short-term rainfall parameters obtained through interpolation, the threshold
spatial distribution for the study area can be derived. Taking H12/D7 as an example, the
process is as follows:
First, by dividing the H12 values of each landslide point by the optimal ratio coefficient
RC12, the corresponding D7 thresholds for each landslide point can be calculated. These D7
thresholds serve as a basis for applying the Kriging interpolation method to obtain the spatial
distribution map of the D7 thresholds across the entire study area.
Next, by multiplying the D7 values of each landslide point by the ratio coefficient RC12,
the corresponding H12 thresholds for each landslide point can be determined. Subsequently,
utilizing these H12 thresholds, the Kriging interpolation method is applied once more to
generate the spatial distribution map of the H12 thresholds for the entire study area.
## 3.3   Typhoon-specific rainfall-induced landslide warning system
In order to effectively prevent typhoon-specific rainfall-induced landslide hazards,
constructing a comprehensive landslide warning system is crucial. This system integrates LSP
with critical rainfall thresholds, combining spatial probability and temporal probability to
predict the risk of landslide occurrence and the timing of potential events.
### 3.3.1   Construction of the landslide warning system

Using the natural breaks point method, the LSP is categorized into five levels of spatial

probability: very low (S1), low (S2), moderate (S3), high (S4), and very high (S5). These
levels represent varying degrees of susceptibility to landslides in different regions, forming
the basis for assessing landslide risks when combined with rainfall data. Paralleling the LSP
categorization, rainfall thresholds are also divided into five levels using the natural breaks
point method, representing temporal probability: very low (T1), low (T2), moderate (T3),
high (T4), and very high (T5). A lower rainfall threshold indicates a higher likelihood of
typhoon-induced landslides, thus signaling a greater risk of landslide events.

**Table 2** Classification of landslide hazard warning zones by integrating landslide susceptibility levels
(S1~S5) with rainfall threshold levels (T1~T5).

| Landslide hazard warning zones | T1 | T2 | T3 | T4 | T5 |
|---|---|---|---|---|---|
| S1 (very low) | No warning zone (2nd level) | No warning zone (1st level) | No warning zone (1st level) | No warning zone (1st level) | No warning zone (1st level) |
| S2 (low) | 3rd level warning zone | No warning zone (2nd level) | No warning zone (2nd level) | No warning zone (1st level) | No warning zone (1st level) |
| S3 (moderate) | 4th level warning zone | 3rd level warning zone | 3rd level warning zone | No warning zone (2nd level) | No warning zone (1st level) |
| S4 (high) | 5th level warning zone | 4th level warning zone | 3rd level warning zone | No warning zone (2nd level) | No warning zone (1st level) |
| S5 (very high) | 5th level warning zone | 5th level warning zone | 4th level warning zone | 3rd level warning zone | No warning zone (2nd level) |

The matrix-based integration of LSP results and rainfall thresholds, as presented in Table

2 (Segoni et al., 2015), highlights the correlation between landslide susceptibility and rainfall
intensity. As the levels of landslide hazard warnings escalate from the 1st level, indicating no
warning, to the 5th level, which signifies the highest alert, the likelihood of landslide
occurrences correspondingly increases. Areas categorized in higher hazard zones correspond
to regions with a heightened risk of landslides. This hazard warning system provides a spatial
framework for risk assessment and early warning, generating hazard zonation maps that can
be integrated into operational landslide monitoring and warning protocols. This underscores
the importance of implementing more effective geological disaster prevention strategies, as
thoroughly discussed in the literature by Huang et al. (2022).

**4 Landslide susceptibility prediction using machine learning models**

**4.1 Statistical analysis of conditioning factors**

The statistical analysis reveals distinct patterns of landslide susceptibility across all conditioning factors (Table S1 in the Supplement). Topographic factors demonstrate clear elevation-dependent behavior, with maximum susceptibility occurring at intermediate elevations (545–782 m, FR=1.637, IV=0.389), suggesting optimal conditions where weathering processes and slope instability converge. Slope gradient exhibits peak susceptibility in the moderate range (7.87–15.06°, FR=1.522, IV=0.343), indicating insufficient driving forces at gentler slopes and potential debris removal at steeper gradients. South-facing aspects show enhanced susceptibility (FR=1.299, IV=0.230), likely attributable to intensified weathering from solar radiation and moisture cycles.

Morphological indices reveal significant correlations with landslide occurrence. Profile curvature demonstrates highest susceptibility in convex areas (0.17–0.59, FR=1.480, IV=0.480), where stress concentration promotes slope failure. TWI shows strong positive correlation with wetness, peaking at high values (8.69–13.62, FR=1.799, IV=0.444), confirming the critical role of water accumulation in slope destabilization. SPI indicates maximum susceptibility in moderate stream power ranges (1.27–2.39, FR=1.298, IV=0.229), reflecting optimal erosional conditions.

Proximity factors exhibit contrasting patterns based on infrastructure type. Distance to roads shows strong inverse correlation with landslide occurrence (0–800 m, FR=1.499, IV=0.333), indicating anthropogenic disturbance effects. Conversely, distance to faults reveals a bimodal pattern with peak susceptibility at intermediate distances (7–12 km, FR=1.439, IV=0.305), suggesting regional structural influence rather than localized fault-induced instability. Environmental factors demonstrate vegetation's protective role, with moderate NDVI values (0.64–0.76) showing elevated susceptibility (FR=1.854, IV=0.015),

representing the transition zone between bare soil vulnerability and established vegetation
stability. Lithological analysis reveals pronounced material control, with rhyolite (FR=1.546,
IV=0.353) and granite (FR=1.247, IV=0.198) showing enhanced susceptibility due to
intensive weathering and joint development, while sedimentary rocks (slate, shale, limestone,
sandstone) exhibit strong resistance (FR<0.21) owing to their structural integrity and lower
weathering susceptibility.
**4.2 Landslide susceptibility modeling in Zixing City**
Prior to model development, multicollinearity analysis was conducted using variance
inflation factor (VIF) to ensure statistical reliability of the conditioning factors. The analysis
revealed method-specific multicollinearity patterns: IV and CF methods showed no
significant multicollinearity issues (all VIF < 10), while the FR method exhibited
multicollinearity in four variables (SPI, Aspect, Plan curvature, and Distance to rivers with
VIF > 10), which were subsequently excluded from FR-based modeling (Table S2 in the
supplement). Following this preprocessing, landslide susceptibility prediction was performed
using SVM and LightGBM models with the three distinct weighting methods (IV, CF, and
FR). Susceptibility levels were categorized into five classes using the natural breaks
classification method, with non-landslide samples strategically selected by excluding buffer
zones of varying distances (0.1, 0.5, 1.0, 2.0, and 5.0 km) around documented landslide
locations to optimize model performance and reduce spatial bias.
**4.2.1 IV-based modeling performance**
The IV-derived susceptibility maps (Fig. 4) revealed distinct spatial patterns between the
two models across varying buffer distances. At smaller scales, the SVM model demonstrated
more detailed classification, with a higher degree of overlap between high susceptibility areas
and actual landslide locations. The LightGBM model's classification was smoother, with a
lower degree of overlap between high susceptibility areas and actual landslide locations.
Notably, this performance discrepancy diminished progressively with increasing buffer
distances.

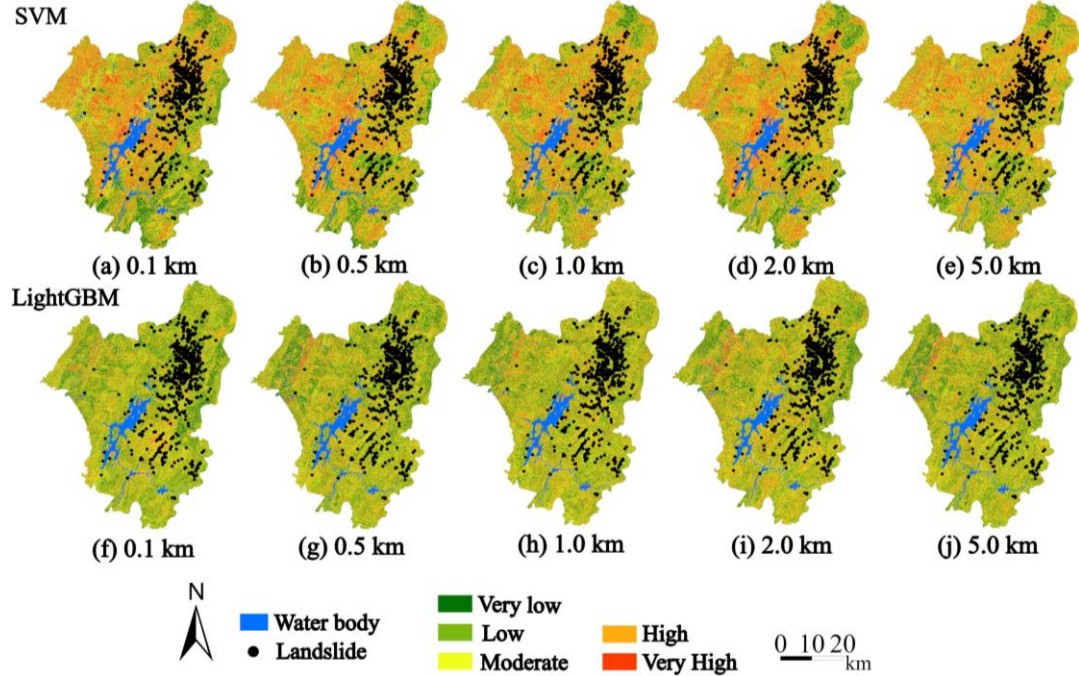

**Figure 4** Landslide susceptibility map based on SVM and LightGBM models using the IV input.

**4.2.2   CF-based modeling performance**
In CF-based modeling (Fig. 5), the SVM model's high and very high landslide
susceptibility areas at smaller scales were more extensive than in the IV mode, with actual
landslide locations more frequently distributed within these high-risk areas. As the scale
increased, the high susceptibility areas gradually decreased. The LightGBM model also
showed a relatively smooth distribution, with some high susceptibility areas identified at
smaller scales gradually integrating as the scale increased, following a similar trend to the
SVM model.

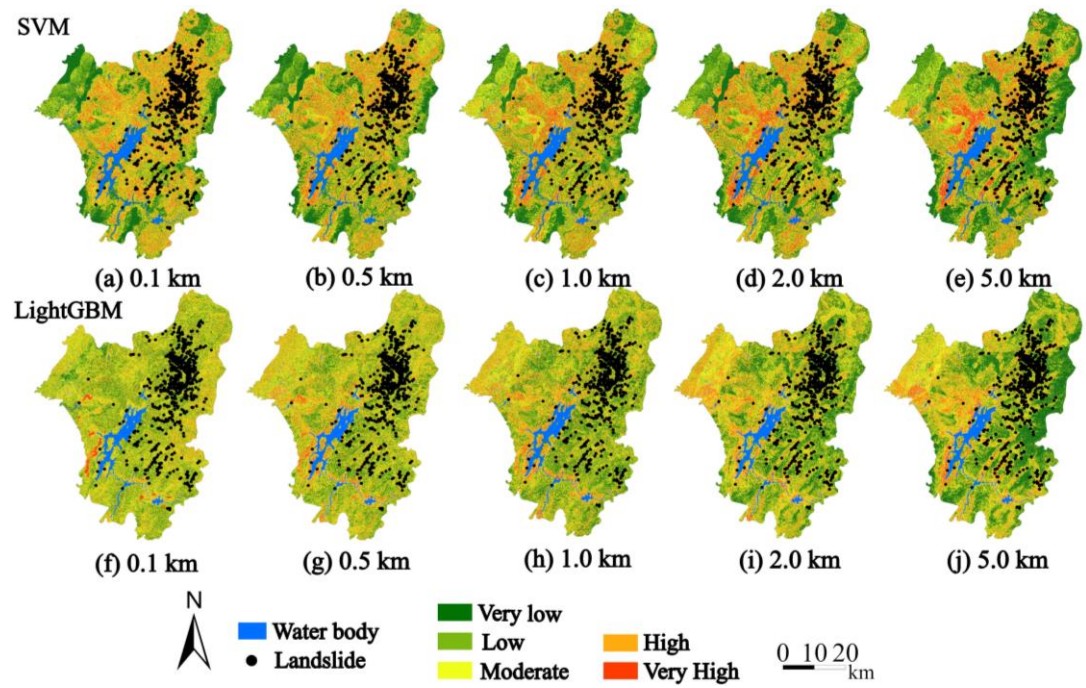

**Figure 5** Landslide susceptibility map based on SVM and LightGBM models using the CF input.

### 4.2.3  FR-based modeling performance

Regarding the FR input (Fig. 6), the SVM model identified a significant number of high and very high landslide susceptibility areas at smaller scales compared to the IV and CF inputs, which closely matched the actual locations of landslides. As the buffer scale expanded, these high-risk areas generally diminished and the distribution became smoother. Conversely, the LightGBM model delivered more uniform results, offering broader moderate-risk distributions, with a small number of high susceptibility areas that did not align with the actual landslide locations. As the scale increased, the high susceptibility areas identified by the LightGBM model gradually diminished, showing greater consistency with the SVM model results at the higher scale.

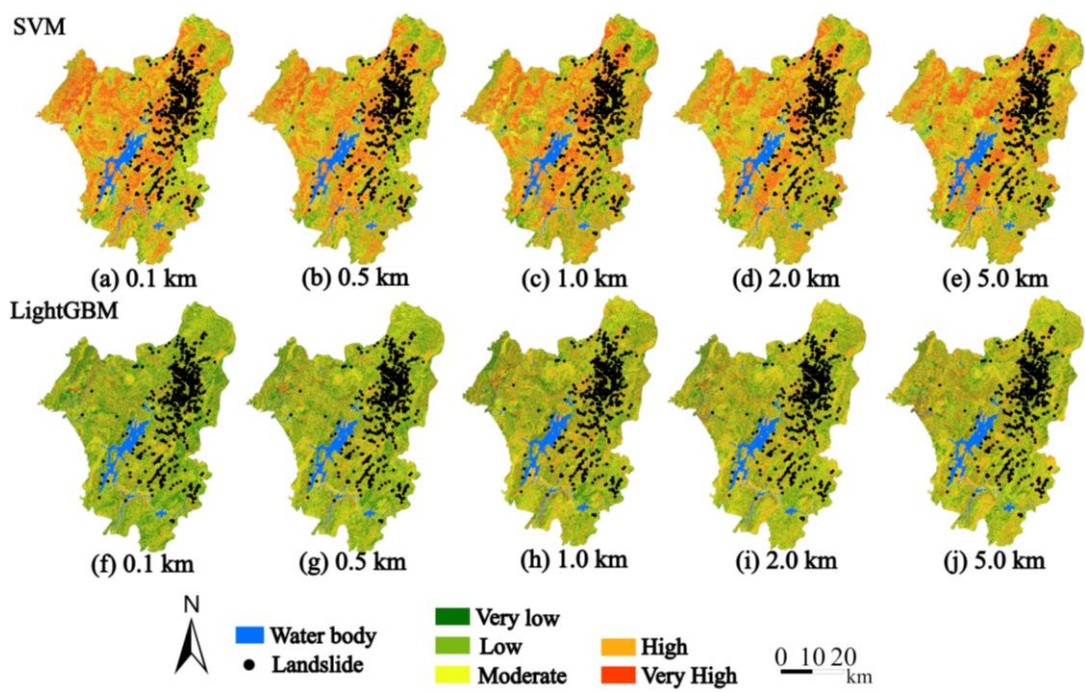

**Figure 6** Landslide susceptibility map based on SVM and LightGBM models using the FR input.

### 4.3 Uncertainty analysis of LSP results

#### 4.3.1 LSP accuracy evaluation and comparative performance

Table S2 (in the Supplement) demonstrates contrasting performance characteristics between the two machine learning approaches across different spatial scales and input configurations. LightGBM consistently achieved high AUC values (0.915–0.921) and maintained stable F1-scores (0.838–0.850) across all buffer distances and input methods, indicating robust generalization capability. In contrast, SVM exhibited pronounced sensitivity to parameter combinations, with performance varying significantly across different buffer distances (F1-scores ranging from 0.681 to 0.859) and input methods, particularly showing notable degradation with FR input at extreme spatial scales (0.1 km and 5.0 km).

Two configurations emerged as comprehensively superior: SVM with FR input at 0.5 km and 2.0 km buffer distances, both achieving F1-scores of 0.859. These optimal configurations not only maintained competitive AUC values (0.914 and 0.913 respectively) but demonstrated superior precision-recall balance compared to corresponding LightGBM configurations (F1-scores: 0.854 and 0.856). The high recall values (0.845 and 0.851) coupled with robust

precision (0.873 and 0.867) indicate enhanced sensitivity to landslide-prone areas while
minimizing false positive predictions. This bimodal performance pattern suggests that
intermediate buffer distances effectively capture fault-related geomorphological processes
influencing slope stability.
Independent validation on the test set confirmed the robustness of these optimal
configurations, with SVM-FR models at 0.5 km and 2.0 km buffer distances achieving F1-
scores of 0.847 and 0.852 respectively, representing minimal performance degradation from
training results. The consistent AUC values (0.909 and 0.908) on the test set further validate
the models' discriminative capability and indicate absence of overfitting, confirming the
reliability of these configurations for practical landslide susceptibility assessment applications.
**4.3.2   LSP distribution characteristics across conditions**
In addition to the performance metrics, the distribution characteristics of landslide
susceptibility predictions revealed fundamental differences between the models (Figs. S1–S3
in the Supplement). LightGBM generated smoother, more symmetrical distributions with
lower mean susceptibility values (0.196–0.320) and smaller standard deviations (0.099–
0.187), indicating stable and uniform predictions. In contrast, SVM exhibited greater
variability, with irregular distributions, higher mean values (0.303–0.515), and larger standard
deviations (0.112–0.214). Notably, SVM's mean susceptibility under FR input rose sharply
(0.446–0.515), while LightGBM maintained lower means despite moderately broader
deviations (0.160–0.187).
Therefore, SVM is preferable for FR-based modeling at 0.5 km and 2.0 km buffers,
where spatial precision is prioritized over prediction uniformity. The SVM model achieved its
highest accuracy at the 0.5 km buffer, classifying 86.4% of recorded landslides in high and
very high susceptibility zones (Fig. 6b). At the 2.0 km buffer (Fig. 6d), it still correctly

classified 82.1% of landslides in these zones. As a result, Fig. 6b is selected as the final

landslide susceptibility map.

## 5 Landslide risk assessment in Zixing City

### 5.1 Critical rainfall thresholds for landslides in Zixing City

We evaluated four rainfall threshold models (H1-D7, H12-D7, H24-D7, and H72-D7)

through 5-fold cross-validation, with their optimal ratio coefficient (RC) thresholds and

prediction accuracies summarized in Table 3. The H24-D7 model, coupling 24-hour rainfall

during landfall with 7-day antecedent moisture, achieved the highest accuracy (71.8%) by

effectively capturing both cumulative saturation and abrupt triggering by typhoon rainfall

bursts. Notably, the H24-D7 model exhibited stable performance across all folds, with

accuracy ranging narrowly between 68.8% (Fold 1) and 74.6% (Fold 4), reflecting robust

generalizability.

**Table 3** Optimal RC values and prediction accuracies (%) for each model across 5-fold cross validation.

| Model | Fold 1 RC/Accuracy | Fold 2 RC/Accuracy | Fold 3 RC/Accuracy | Fold 4 RC/Accuracy | Fold 5 RC/Accuracy | Average RC/Accuracy |
|---|---|---|---|---|---|---|
| H1-D7 | 0.032/56.5 | 0.062/29.7 | 0.076/35.5 | 0.022/53.6 | 0.040/47.8 | 0.047/44.6 |
| H12-D7 | 0.077/54.2 | 0.167/46.6 | 0.243/48.3 | 0.267/47.7 | 0.154/45.3 | 0.182/48.5 |
| H24-D7 | 0.472/68.8 | 0.436/72.3 | 0.422/73.1 | 0.459/74.6 | 0.414/70.2 | **0.440/71.8** |
| H72-D7 | 0.789/56.5 | 0.776/59.4 | 0.781/63.1 | 0.802/51.4 | 0.783/60.1 | 0.787/58.1 |

In contrast, the H1-D7 and H12-D7 models displayed marked instability: H1-D7

accuracy fluctuated between 29.7% (Fold 2) and 56.5% (Fold 1), while H12-D7 thresholds

(RC12: 0.077–0.267) corresponded to accuracies of 45.3–48.3%. The H72-D7 model showed

moderate performance variability (accuracy: 51.4–63.1%) despite consistently high RC72

thresholds (>0.78).

These results highlight the critical role of temporal rainfall parameter selection. The

superior performance of the H24-D7 model (24-hour short-term rainfall and 7-day antecedent

rainfall) suggests that a 24-hour duration optimally captures both immediate landslide triggers

and cumulative hydrological effects, balancing sensitivity and stability. Shorter (H1/H12) or longer (H72) durations either overemphasize transient rainfall spikes or dilute critical triggering signals.

## 5.2 Spatio-temporal distribution of rainfall thresholds

Fig. 7 illustrates the spatial distribution of rainfall-triggered landslide thresholds derived from four models (RC1, RC12, RC24, and RC72) across multiple temporal scales (1-hour, 12-hour, 24-hour, 72-hour, and 7-day) within the study area.

### 5.2.1 Short-term predictions (1-hour to 12-hour scales)

At the 1-hour scale (Fig. 7a), the RC1 model generated thresholds ranging from 7 to 50 mm, with 65.2% of landslides occurring in moderate threshold zones (20–30 mm). This indicates the model's effectiveness in detecting slope failures under short-duration rainfall. In contrast, the RC12 model on the 12-hour scale (Fig. 7b) showed a wider threshold range (25–200 mm), with 62.9% of landslides in mid-to-high threshold regions (80–130 mm). This mismatch suggests that the 12-hour cumulative data may underestimate rainfall impacts in specific topographic settings.

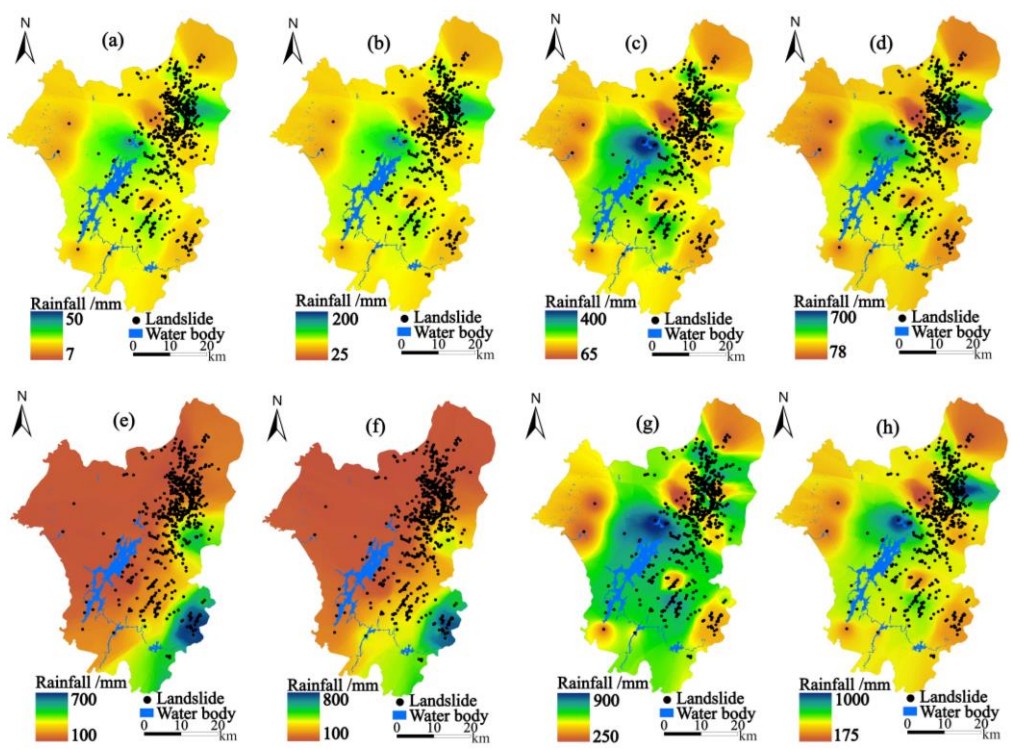

**Figure 7** Distribution of typhoon rainfall thresholds under various optimal RC ratios: (a) 1-hour RC1-based, (b) 12-hour RC12-based, (c) 24-hour RC24-based, (d) 72-hour RC72-based, (e) 7-day RC1-based, (f) 7-day RC12-based, (g) 7-day RC24-based, and (h) 7-day RC72-based.

### 5.2.2    Mid-term predictions (24-hour to 72-hour scales)

The RC24 model at the 24-hour scale (Fig. 7c) displayed a threshold range of 65–400 mm, with 87.1% of landslides occurring within moderate thresholds (100–250 mm) and 12.3% in higher thresholds (>250 mm). This indicates a more accurate capture of rainfall intensity effects. At the 72-hour scale (Fig. 7d), the RC72 model produced thresholds between 78–700 mm, with 59.2% of landslides in mid-to-high threshold regions (200–500 mm). Although the RC72 model demonstrated reasonable sensitivity to prolonged rainfall, its upper threshold (700 mm) may result in conservative risk predictions for some geological settings.

### 5.2.3    Long-term predictions (7-day scale)

At the 7-day scale, significant differences emerge across models in terms of predicted rainfall thresholds and landslide points. The RC1 model (Fig. 7e) shows a threshold range of 100–700 mm, with landslide points predominantly concentrated in the lower rainfall ranges. While these low-threshold landslides may indicate localized risks, the model's conservative threshold distribution fails to effectively capture landslides triggered by higher rainfall amounts, potentially overlooking more significant events.

The RC12 model (Fig. 7f), with a threshold range of 100–800 mm, also shows a concentration of landslide points in the lower rainfall ranges. Despite a wider threshold range, the similarity to the RC1 model suggests that RC12 may also underutilize its capacity to predict higher typhoon-induced landslides, leading to under-prediction in areas experiencing moderate to heavy precipitation.

In contrast, the RC24 model (Fig. 7g) exhibits a balanced threshold range (250–900 mm) and effectively identifies landslide points in both moderate and high rainfall categories. This balance enables RC24 to capture the full spectrum of typhoon-induced landslides, accurately identifying risks across different rainfall intensities.

The RC72 model (Fig. 7h) shows a concentration of landslide points in the higher
rainfall range (175–1000 mm). While it predicts landslides accurately under heavy rainfall
conditions, the model may overestimate risks in some regions and neglect potential landslides
associated with lower rainfall thresholds.
Based on the above analysis, the RC24 model is the optimal choice, which aligns with
the findings in Section 5.1. Its effectiveness is evident as it demonstrates superior stability and
accuracy in both the 24-hour and 7-day timescales.The RC24 model's balanced threshold
range allows it to accurately assess landslide risks across varying rainfall intensities. This
makes it the most reliable choice for practical landslide hazard warning applications.
**5.3    Landslide hazard warning system for Zixing City**
Based on the optimal LSP results (Fig. 6b) and the validated RC24 rainfall threshold
model, a spatially explicit landslide hazard warning system was established for Zixing City.
The integration of spatial probability (LSP) and temporal probability (rainfall thresholds)
followed the matrix classification outlined in Table 2.

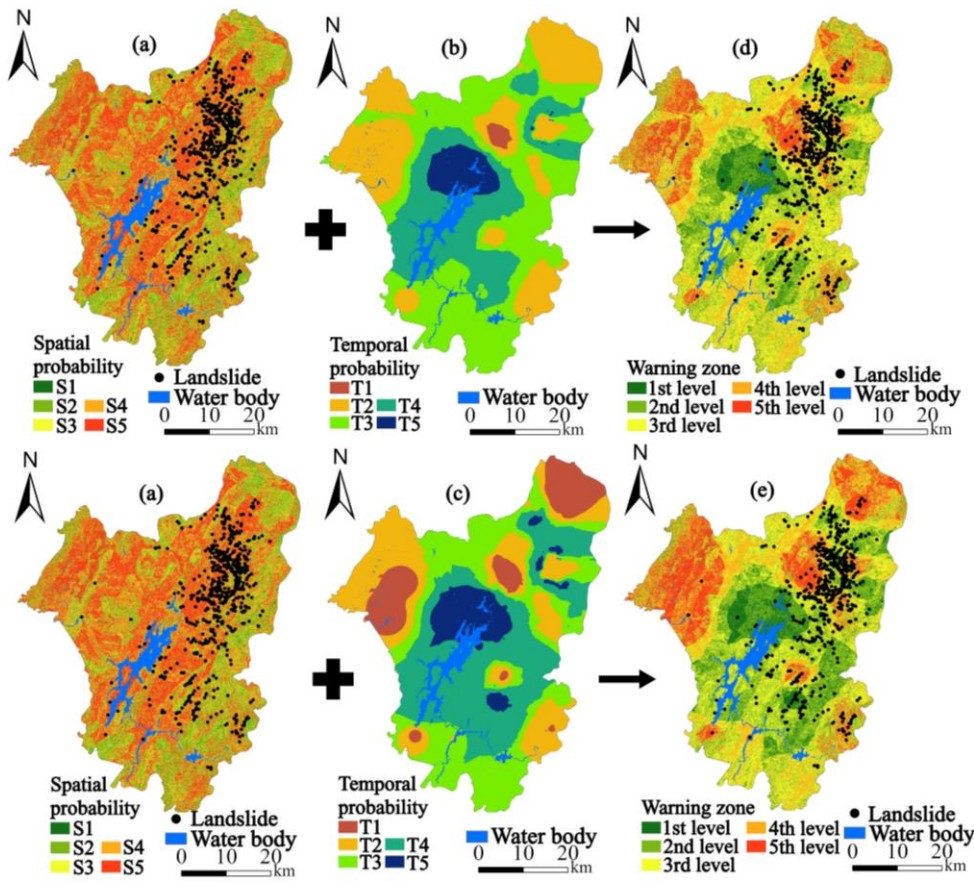

**Figure 8** Landslide warning zones generated by overlaying spatial and temporal probability maps: (a) optimal spatial probability, (b) 24-hour RC24-based rainfall threshold, (c) 7-day RC24-based rainfall threshold, (d) overlay of (a) and (b), and (e) overlay of (a) and (c).

Five susceptibility levels in the LSP map (Fig. 6b) were replaced with five spatial probabilities (S1–S5) (Fig. 8a), respectively. Simultaneously, the spatially interpolated 24-hour rainfall thresholds (H24) (Fig. 8b) and 7-day effective rainfall thresholds (D7) (Fig. 8c) derived from the RC24 model were classified into five temporal probability levels (T1–T5) using the natural breaks method. Spatial overlay analysis was performed to combine the susceptibility levels (S1–S5) with the rainfall threshold levels (T1–T5), generating two hazard warning zone maps: H24-based (Fig. 8d) and D7-based (Fig. 8e).

Quantitative assessment of both warning systems reveals distinct performance characteristics. The 24-hour threshold system (Fig. 8d) demonstrates superior predictive efficiency, with 71.4% of historical landslides occurring within high to very high warning zones (Levels 3–5) while covering only 34.2% of the total area, resulting in an efficiency ratio

of 2.09 and a risk density of 49.0 landslides per 1000 high-risk grid cells. The spatial
distribution shows concentrated high-risk areas primarily in the central region, characterized
by steep slopes (>21.80°), weathered granite lithology, and road proximity (0–800 m). This
focused distribution indicates effective identification of areas most sensitive to short-term
intense rainfall triggers.
The 7-day threshold system (Fig. 8e) exhibits broader spatial coverage, with high-risk
zones encompassing 42.7% of the study area and capturing 68.7% of historical landslides,
yielding a lower efficiency ratio of 1.61 and risk density of 37.8 landslides per 1000 grid cells.
This system effectively identifies extended vulnerable areas in northern and eastern regions,
reflecting cumulative rainfall effects on slope stability. The expanded coverage captures zones
where prolonged antecedent moisture interacts with moderate-to-high susceptibility
conditions.
Statistical validation confirms the complementary nature of both systems. The 24-hour
system achieves higher spatial efficiency (efficiency ratio 2.09 vs. 1.61) and landslide
concentration (risk density 49.0 vs. 37.8), making it optimal for immediate typhoon response
and targeted emergency resource allocation. Conversely, the 7-day system provides
comprehensive coverage for prolonged rainfall scenarios, essential for early warning during
extended typhoon events despite its broader spatial distribution and lower concentration
efficiency. The combined application of both systems enables dynamic hazard assessment,
addressing both rapid-onset failures during typhoon landfall and delayed failures following
sustained precipitation.
**6   Discussion**
**6.1   Model selection strategy and optimization of LSP**

Our comparative analysis of SVM and LightGBM across different input methods (IV,
CF, FR) and buffer distances shows distinct performance patterns crucial for model selection
in typhoon-induced LSP. SVM exhibited marked sensitivity to configuration parameters, with
F1-scores varying from 0.681 to 0.859 depending on buffer distance and input method.
LightGBM maintained more stable performance (F1-scores: 0.838–0.850) across all
configurations. These differences reflect fundamental algorithmic characteristics: SVM's
kernel-based approach effectively captures localized patterns when properly tuned, while
LightGBM's ensemble structure delivers consistent results across varying data conditions.
SVM's superior performance at 0.5–2.0 km buffer distances with FR weighting builds on
findings by Kalantar et al. (2018) and Bogaard and Greco (2018). This buffer range appears
effective for capturing the spatial patterns of typhoon-induced failures in our study area. FR
weighting's effectiveness supports Reichenbach et al. (2018) and Yan et al. (2019), who found
that frequency-based methods excel at quantifying terrain-landslide relationships. In typhoon
conditions, FR effectively weights critical factors including road proximity and weathered
granite lithology.
These performance patterns justify our dual-model approach. SVM, though requiring
careful calibration, enables precise delineation of high-risk zones essential for emergency
response, with SVM-FR at 0.5 km achieving peak accuracy (F1=0.859). LightGBM's
robustness suits operational contexts requiring consistent predictions under variable
conditions. Our results suggest that effective model selection depends on matching
algorithmic strengths to specific application requirements rather than identifying a universally
superior algorithm.
**6.2 Rainfall threshold modeling and typhoon-specific mechanisms**

The H24-D7 model achieved 71.8% accuracy, outperforming alternative temporal
windows (Table 3). The optimal RC24 value of 0.440 (with inter-fold variation of 0.414–
0.472) indicates that landslides typically occur when 24-hour rainfall constitutes
approximately 44% of the preceding 7-day accumulation. This pattern is consistent with the
multi-temporal triggering framework proposed by Nolasco-Javier and Kumar (2018) for
typhoon contexts, where both antecedent saturation and short-term intensity contribute to
slope failure. However, the specific hydrological mechanisms underlying this ratio require
verification through in-situ soil moisture monitoring. The H1-D7 and H12-D7 models showed
lower and more variable accuracy (44.6% and 48.5% respectively), suggesting that shorter
accumulation periods may inadequately represent the cumulative soil saturation process
relevant to this region's geological conditions (Kirschbaum and Stanley, 2018).
Spatial patterns in rainfall thresholds reveal systematic variations across the study area.
Southeastern regions exhibit elevated H24 thresholds exceeding 250 mm (Fig. 7c), while
northern areas show reduced thresholds of 100–150 mm. These spatial variations align with
findings by Lee et al. (2018) and Cho et al. (2022) regarding topographic controls on
typhoon-induced landslides, though the specific mechanisms require further investigation
with detailed meteorological analysis. The spatially distributed thresholds derived through
Kriging interpolation (Table 1) provide location-specific values that improve upon uniform
regional thresholds typically employed in operational systems (Segoni et al., 2018b).
The consistent performance across the five validation folds (68.8–74.6% accuracy)
demonstrates the model's stability when applied to different spatial subsets of the landslide
inventory. This suggests the H24-D7 relationship captures generalizable rainfall-slope
response patterns rather than site-specific anomalies, though validation with independent
typhoon events would further confirm model robustness.
**6.3    Integration of susceptibility and rainfall thresholds for landslide warning**
The integrated warning system combines static susceptibility surfaces with spatially
continuous rainfall thresholds following the matrix framework in Table 2. The H24-based
system (Fig. 8d) captured 71.4% of historical landslides within high to very high warning
zones (Levels 3–5) covering 34.2% of the study area, yielding an efficiency ratio of 2.09. The
D7-based system (Fig. 8e) identified 68.7% of landslides across 42.7% of the area (efficiency
ratio: 1.61). These focused distributions contrast with the broader spatial coverage typically
required by uniform regional thresholds (Guzzetti et al., 2020), though direct comparative
validation would be needed to quantify the performance gain.
The dual-threshold configuration provides complementary perspectives suited to
different phases of typhoon evolution, with D7 reflecting cumulative moisture conditions and
H24 capturing immediate triggering rainfall. This combination addresses the compound
rainfall mechanisms documented in typhoon-affected regions (Gariano et al., 2015; Nolasco-
Javier and Kumar, 2018), though the optimal application strategy for operational warning
would require integration with real-time meteorological forecasting systems.
Spatially continuous thresholds (Fig. 8b, c) address terrain-induced variability more
effectively than point-based approaches. The Kriging interpolation method provides threshold
estimates across the entire study area, accounting for spatial autocorrelation in rainfall
patterns (Table 1). However, threshold accuracy depends on rain gauge density and may
decline in areas distant from monitoring stations, as indicated by the interpolation validation
metrics (R: 0.76–0.87, NSE: 0.71–0.82). The framework advances beyond existing point-
based threshold systems (Segoni et al., 2018b; Guzzetti et al., 2020) by providing spatially
explicit hazard assessment, though regional adaptation of threshold parameters would be
necessary for application in different geological settings.
The modular design allows the framework to be adapted for operational landslide early
warning, though practical implementation would require integration with meteorological

694 monitoring infrastructure, standardized protocols for warning dissemination, and post-event

695 validation procedures to maintain system reliability. These operational considerations extend

696 beyond the methodological scope of this study but represent important directions for future

697 development of typhoon-specific landslide warning systems.

698 **6.4 Limitations and future research directions**

699 Despite promising advancements, this study has limitations owing to the complexity of

700 typhoon-induced landslides. First, the model's validation relies solely on landslides from

701 Typhoon Gaemi. While this single event provided a comprehensive dataset, validating against

702 multiple, varied typhoons is crucial for model robustness. Typhoons differ significantly in

703 intensity, rainfall patterns, forward speed, and seasonality, all of which can influence

704 threshold parameters. For instance, a slow-moving typhoon with higher cumulative rainfall

705 and lower peak intensity could alter the optimal H24-D7 ratios. Future research should

706 incorporate landslide inventories from typhoons with contrasting characteristics to assess

707 threshold transferability and develop adaptive parameterization. The framework's modular

708 design readily facilitates this by allowing recalibration of the RC24 coefficient for different

709 typhoon types.

710 Second, the current study primarily addresses rainfall-induced landslides, overlooking

711 other potential contributing factors. Future work should explore integrating multiple

712 triggering mechanisms, including earthquakes, human-induced slope modifications, and

713 typhoon rainfall, for a more comprehensive hazard assessment.

714 Third, the study doesn't explicitly address the potential impacts of climate change on

715 typhoon rainfall and landslide occurrence. As climate change alters typhoon frequency,

716 intensity, and tracks, future studies should incorporate climate projections specific to

717 typhoon-prone regions. This will enable the development of forward-looking landslide

718 warning systems that can adapt to the evolving threats posed by typhoon-specific rainfall.

Fourth, while this study demonstrates the effectiveness of ML approaches, further refinement is possible. Future research should explore advanced deep learning techniques and ensemble methods to better capture the complex, non-linear relationships between typhoon-related variables (e.g., rainfall intensity, duration, antecedent moisture) and slope stability. These advanced methods may offer improved predictive accuracy, more robust uncertainty quantification, and ultimately, more reliable hazard warnings.

Finally, climate projections for Southeast China show a 15–25% increase in peak typhoon rainfall by 2080 (RCP8.5), which could alter the H24–D7 landslide thresholds from this study. Higher atmospheric moisture may lower D7 thresholds, while greater rainfall intensity could require new H24 parameters. Shifting typhoon tracks and seasonality might also change which areas are vulnerable. Future work must use downscaled climate data to create non-stationary thresholds, ensuring the long-term reliability of warning systems in the region.

**7 Conclusions**

This study establishes a novel integrated framework combining optimized LSP with typhoon-specific rainfall threshold modeling for comprehensive hazard assessment in mountainous regions. Through systematic analysis of 705 landslides triggered by Typhoon Gaemi in Zixing City, several key insights emerge:

(1) Buffer distance optimization proves critical for typhoon-induced landslide modeling, with SVM-FR combinations at 0.5–2.0 km distances achieving superior performance (F1-score: 0.859) compared to conventional approaches. This spatial scale effectively captures typhoon-induced moisture infiltration patterns that differ fundamentally from other triggering mechanisms.

(2) The H24-D7 threshold model demonstrates exceptional stability (71.8% accuracy across 5-fold validation), successfully characterizing the dual-phase failure mechanism unique

to typhoons: prolonged antecedent saturation coupled with intense precipitation bursts during
typhoon passage.
(3) Spatially distributed rainfall thresholds reveal significant heterogeneity, reflecting
complex interactions between typhoon structure and local topography that contradict uniform
regional threshold assumptions in existing operational systems.
(4) The integrated warning system achieves operational efficiency through dual-
threshold configuration: H24 thresholds provide immediate response capability during
typhoon landfall, while D7 thresholds enable early detection of vulnerable areas approaching
saturation conditions.
(5) This framework addresses three critical gaps in current landslide prediction:
systematic buffer optimization for imbalanced datasets, effective integration of variable
weighting with machine learning algorithms, and development of typhoon-specific spatially
explicit thresholds.


*Code and data availability:* The source code and data will be made available on request.
*Competing interests.* The contact author has declared that none of the authors has any
competing interests.
***Author contributions:*** **Weifeng Xiao**: Writing-review & editing, Validation,
Conceptualization. **Guangchong Yao**: Visualization, Validation, Data curation. **Zhenghui**
**Xiao**: Writing-review & editing, Formal analysis. **Ge Liu**: Correspondence, Funding
acquisition. **Luguang Luo**: Visualization, Validation, Investigation, Data curation. **Yunjiang**
**Cao**: Visualization, Formal analysis, Data curation. **Wei Yin**: Validation, Investigation.
**Acknowledgments**

This research was jointly funded by the National Key Research and Development Program of China (2024YFD1501100 and 2024YFD1500602), the National Natural Science Foundation of China (No. 42171385, U2243230), the Key Research Project on High-Efficiency and Green Agricultural Production Technologies in Jilin Province (20230202040NC)and the Youth Innovation Promotion Association of Chinese Academy of Sciences, China (2022228).

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

**Tables**

**Table 1** Kriging interpolation accuracy assessment for rainfall parameters.

| Parameter | RMSE (mm) | MAE | R | NSE |
|---|---|---|---|---|
| H1 | 4.2 | 3.1 | 0.76 | 0.71 |
| H12 | 11.7 | 8.9 | 0.83 | 0.78 |
| H24 | 16.3 | 12.6 | 0.87 | 0.82 |
| H72 | 24.8 | 18.4 | 0.81 | 0.77 |
| D7 | 29.6 | 22.7 | 0.78 | 0.73 |

**Table 2** Classification of landslide hazard warning zones by integrating landslide susceptibility levels (S1~S5) with rainfall threshold levels (T1~T5).

| Landslide hazard warning zones | T1 | T2 | T3 | T4 | T5 |
|---|---|---|---|---|---|
| S1 (very low) | No warning zone (2$^{nd}$ level) | No warning zone (1$^{st}$ level) | No warning zone (1$^{st}$ level) | No warning zone (1$^{st}$ level) | No warning zone (1$^{st}$ level) |
| S2 (low) | 3$^{rd}$ level warning zone | No warning zone (2$^{nd}$ level) | No warning zone (2$^{nd}$ level) | No warning zone (1$^{st}$ level) | No warning zone (1$^{st}$ level) |
| S3 (moderate) | 4$^{th}$ level warning zone | 3$^{rd}$ level warning zone | 3$^{rd}$ level warning zone | No warning zone (2$^{nd}$ level) | No warning zone (1$^{st}$ level) |
| S4 (high) | 5$^{th}$ level warning zone | 4$^{th}$ level warning zone | 3$^{rd}$ level warning zone | No warning zone (2$^{nd}$ level) | No warning zone (1$^{st}$ level) |
| S5 (very high) | 5$^{th}$ level warning zone | 5$^{th}$ level warning zone | 4$^{th}$ level warning zone | 3$^{rd}$ level warning zone | No warning zone (2$^{nd}$ level) |

**Table 3** Optimal RC values and prediction accuracies (%) for each model across 5-fold cross validation.

| Model | Fold 1 RC/Accuracy | Fold 2 RC/Accuracy | Fold 3 RC/Accuracy | Fold 4 RC/Accuracy | Fold 5 RC/Accuracy | Average RC/Accuracy |
|---|---|---|---|---|---|---|
| H1-D7 | 0.032/56.5 | 0.062/29.7 | 0.076/35.5 | 0.022/53.6 | 0.040/47.8 | 0.047/44.6 |
| H12-D7 | 0.077/54.2 | 0.167/46.6 | 0.243/48.3 | 0.267/47.7 | 0.154/45.3 | 0.182/48.5 |
| H24-D7 | 0.472/68.8 | 0.436/72.3 | 0.422/73.1 | 0.459/74.6 | 0.414/70.2 | **0.440/71.8** |
| H72-D7 | 0.789/56.5 | 0.776/59.4 | 0.781/63.1 | 0.802/51.4 | 0.783/60.1 | 0.787/58.1 |

**Figure captions**

**Figure 1** Geographical distribution of the study area, landslides and rainfall gauges.

**Figure 2** Landslide-related conditioning factors.

**Figure 3** Technical framework for developing a typhoon-specific rainfall-induced landslide warning system.

**Figure 4** Landslide susceptibility map based on SVM and LightGBM models using the IV input.
**Figure 5** Landslide susceptibility map based on SVM and LightGBM models using the CF input.
**Figure 6** Landslide susceptibility map based on SVM and LightGBM models using the FR input.
**Figure 7** Distribution of typhoon rainfall thresholds under various optimal RC ratios: (a) 1-hour RC1-based,
(b) 12-hour RC12-based, (c) 24-hour RC24-based, (d) 72-hour RC72-based, (e) 7-day RC1-based, (f) 7-day
RC12-based, (g) 7-day RC24-based, and (h) 7-day RC72-based.
**Figure 8** Landslide warning zones generated by overlaying spatial and temporal probability maps: (a) optimal
spatial probability, (b) 24-hour RC24-based rainfall threshold, (c) 7-day RC24-based rainfall threshold, (d)
overlay of (a) and (b), and (e) overlay of (a) and (c).

**Title page**
**Title**: From typhoon rainfall to slope failure: optimizing susceptibility models and dynamic thresholds for
landslide warnings in Zixing City, China
**Authors**:
Weifeng Xiao[1], Guangchong Yao[1], Zhenghui Xiao[1], Ge Liu[*2], Luguang Luo[1], Yunjiang Cao[1], Wei Yin[3]
**Affiliations**:
[1]School of Earth Sciences and Spatial Information Engineering, Hunan University of Science and
Technology, Xiangtan 411201, China
[2]Northeast Institute of Geography and Agroecology, CAS, Changchun 130102, China
[3]Hunan Institute of Geological Disaster Investigation and Monitoring, Changsha 410004, China

**Corresponding author:**
E-mail address: liuge@iga.ac.cn (Ge Liu), Northeast Institute of Geography and Agroecology, CAS,
Changchun 130102, China













**Supplement**

**Table S1** IV, CF and FR values for each conditioning factor.

| Conditioning factors | Factor grading | Landslides | IV | CF | FR |
|---|---|---|---|---|---|
| Elevation (m) | 92~314 | 81 | -0.493 | -0.679 | 0.507 |
| | 314~545 | 255 | 0.218 | 0.246 | 1.279 |
| | 545~782 | 312 | 0.389 | 0.493 | 1.637 |
| | 782~1098 | 57 | -0.505 | -0.704 | 0.495 |
| | 1098~2033 | 0 | -1 | 0 | 0 |
| Slope (°) | 0~7.87 | 91 | -0.347 | -0.427 | 0.653 |
| | 7.87~15.06 | 267 | 0.343 | 0.420 | 1.522 |
| | 15.06~21.80 | 219 | 0.168 | 0.184 | 1.202 |
| | 21.80~29.44 | 112 | -0.213 | -0.240 | 0.786 |
| | 29.44~57.31 | 16 | -0.756 | -1.411 | 0.2440 |
| Aspect | Plan | 0 | -1 | 0 | 0 |
| | North | 74 | -0.102 | -0.109 | 0.897 |
| | Northeast | 67 | -0.058 | -0.060 | 0.942 |
| | East | 70 | -0.120 | -0.128 | 0.800 |
| | Southeast | 105 | 0.116 | 0.123 | 1.131 |
| | South | 102 | 0.230 | 0.261 | 1.299 |
| | Southwest | 96 | 0.144 | 0.156 | 1.169 |
| | West | 96 | 0.039 | 0.039 | 1.040 |
| | Northwest | 95 | -0.071 | -0.074 | 0.929 |
| Plan curvature | -3.73~-0.57 | 36 | -0.275 | -0.321 | 0.725 |
| | -0.57~-0.18 | 189 | 0.250 | 0.287 | 1.333 |
| | -0.18~0.15 | 284 | 0.000 | 0.000 | 1.000 |
| | 0.15~0.54 | 156 | -0.059 | -0.061 | 0.941 |
| | 0.54~3.94 | 40 | 0.373 | -0.467 | 0.627 |

| | | | | | |
|---|---|---|---|---|---|
| | -3.92~-0.55 | 19 | -0.608 | -0.935 | 0.392 |
| | -0.55~-0.16 | 114 | -0.240 | -0.274 | 0.760 |
| Profile curvature | -0.16~0.17 | 260 | -0.112 | -0.119 | 0.888 |
| | 0.17~0.59 | 253 | 0.480 | 0.392 | 1.480 |
| | 0.59~3.76 | 59 | 0.276 | 0.324 | 1.382 |
| | 1.98~4.40 | 151 | -0.393 | -0.499 | 0.607 |
| TWI | 4.40~5.54 | 297 | 0.245 | 0.280 | 1.324 |
| | 5.54~6.91 | 132 | -0.011 | -0.011 | 0.989 |
| | 6.91~8.69 | 73 | 0.046 | 0.047 | 1.048 |
| | 8.69~13.62 | 52 | 0.444 | 0.587 | 1.799 |
| | 0~800 | 350 | 0.333 | 0.405 | 1.499 |
| | 800~2000 | 194 | -0.011 | -0.011 | 0.989 |
| Distance to road (m) | 2000~4500 | 153 | -0.277 | -0.324 | 0.723 |
| | 4500~7500 | 8 | -0.857 | -1.942 | 0.143 |
| | 7500~9700 | 0 | -1.000 | 0.001 | 0.001 |
| | 0~800 | 152 | 0.147 | 0.158 | 1.172 |
| | 800~2200 | 205 | 0.081 | 0.085 | 1.088 |
| Distance to river (m) | 2200~4500 | 218 | 0.010 | 0.010 | 1.010 |
| | 4500~8000 | 101 | -0.229 | -0.260 | 0.771 |
| | 8000~12800 | 29 | -0.278 | -0.325 | 0.722 |
| | 0~2000 | 64 | -0.380 | -0.478 | 0.620 |
| | 2000~7000 | 262 | 0.062 | 0.064 | 1.066 |
| Distance to fault (m) | 7000~12000 | 286 | 0.305 | 0.364 | 1.439 |
| | 12000~18000 | 62 | -0.414 | -0.535 | 0.586 |
| | 18000~28100 | 31 | -0.398 | -0.508 | 0.602 |
| | -0.20~0.27 | 2 | -0.956 | -3.133 | 0.044 |
| | 0.27~0.47 | 29 | -0.446 | -0.591 | 0.554 |
| NDVI | 0.47~0.64 | 108 | 0.217 | 0.245 | 1.278 |
| | 0.64~0.76 | 296 | 0.015 | 0.617 | 1.854 |
| | 0.76~0.94 | 270 | -0.255 | -0.295 | 0.745 |
| | -8.46~-2.72 | 0 | -1.000 | 0.000 | 0.000 |
| | -2.72~1.27 | 108 | 0.250 | 0.288 | 1.334 |
| SPI | 1.27~2.39 | 370 | 0.229 | 0.261 | 1.298 |
| | 2.39~3.46 | 180 | -0.320 | -0.386 | 0.680 |
| | 3.46~7.45 | 47 | -0.356 | -0.440 | 0.644 |

| | Slate | 8 | -0.856 | -1.938 | 0.144 |
|---|---|---|---|---|---|
| | Shale | 10 | -0.798 | -1.601 | 0.202 |
| Lithology | Limestone | 1 | -0.907 | -2.376 | 0.093 |
| | Sandstone | 3 | -0.958 | -3.179 | 0.042 |
| | Granite | 485 | 0.198 | 0.221 | 1.247 |
| | Rhyolite | 198 | 0.353 | 0.436 | 1.546 |

**Table S2** Performance metrics of SVM and LightGBM models across different buffer distances and input methods (Training set).

| Buffer Distance (km) | Model | Input Method | AUC | Precision | Recall | F1-score |
|---|---|---|---|---|---|---|
| 0.1 | SVM | IV | 0.831 | 0.798 | 0.752 | 0.774 |
| | | CF | 0.812 | 0.781 | 0.736 | 0.758 |
| | | FR | 0.720 | 0.695 | 0.668 | 0.681 |
| | LightGBM | IV | 0.919 | 0.964 | 0.823 | 0.843 |
| | | CF | 0.919 | 0.867 | 0.821 | 0.843 |
| | | FR | 0.915 | 0.859 | 0.818 | 0.838 |
| 0.5 | SVM | IV | 0.825 | 0.792 | 0.743 | 0.767 |
| | | CF | 0.820 | 0.786 | 0.739 | 0.762 |
| | | **FR** | **0.914** | **0.873** | **0.845** | **0.859** |
| | LightGBM | IV | 0.920 | 0.866 | 0.825 | 0.845 |
| | | CF | 0.920 | 0.868 | 0.823 | 0.845 |
| | | FR | 0.921 | 0.881 | 0.829 | 0.854 |
| 1.0 | SVM | IV | 0.826 | 0.794 | 0.745 | 0.769 |
| | | CF | 0.819 | 0.783 | 0.741 | 0.761 |
| | | FR | 0.721 | 0.698 | 0.671 | 0.684 |
| | LightGBM | IV | 0.920 | 0.867 | 0.827 | 0.844 |
| | | CF | 0.920 | 0.869 | 0.825 | 0.846 |
| | | FR | 0.916 | 0.861 | 0.822 | 0.841 |
| 2.0 | SVM | IV | 0.826 | 0.795 | 0.747 | 0.770 |
| | | CF | 0.834 | 0.801 | 0.756 | 0.778 |
| | | **FR** | **0.913** | **0.867** | **0.851** | **0.859** |
| | LightGBM | IV | 0.920 | 0.868 | 0.829 | 0.848 |
| | | CF | 0.920 | 0.870 | 0.821 | 0.848 |
| | | FR | 0.918 | 0.882 | 0.831 | 0.856 |
| | | IV | 0.823 | 0.791 | 0.741 | 0.765 |

| | | | | | | |
|---|---|---|---|---|---|---|
| | SVM | CF | 0.883 | 0.843 | 0.798 | 0.820 |
| 5.0 | | FR | 0.721 | 0.697 | 0.669 | 0.683 |
| | | IV | 0.919 | 0.865 | 0.831 | 0.850 |
| | LightGBM | CF | 0.918 | 0.871 | 0.829 | 0.850 |
| | | FR | 0.916 | 0.862 | 0.823 | 0.842 |


**Table S3** Variance Inflation Factor (VIF) analysis for landslide conditioning factors.

| Conditioning factors | IV method | CF method | FR method |
|---|---|---|---|
| Elevation (m) | 2.34 | 2.41 | 2.36 |
| Slope (°) | 3.67 | 3.52 | 3.88 |
| Aspect | 1.89 | 1.94 | **12.45** |
| Profile curvature | 2.15 | 2.08 | 2.33 |
| Plan curvature | 1.76 | 1.82 | **11.23** |
| TWI | 4.23 | 4.18 | 4.11 |
| Distance to road (m) | 3.45 | 3.38 | 4.12 |
| Distance to river (m) | 6.78 | 6.92 | **10.56** |
| Distance to fault (m) | 2.56 | 2.63 | 2.41 |
| NDVI | 2.91 | 2.87 | 3.15 |
| SPI | 5.67 | 5.84 | **13.89** |
| Lithology | 1.98 | 2.05 | 1.87 |

*Note*: VIF values > 10 indicate multicollinearity issues and variables were excluded from analysis.











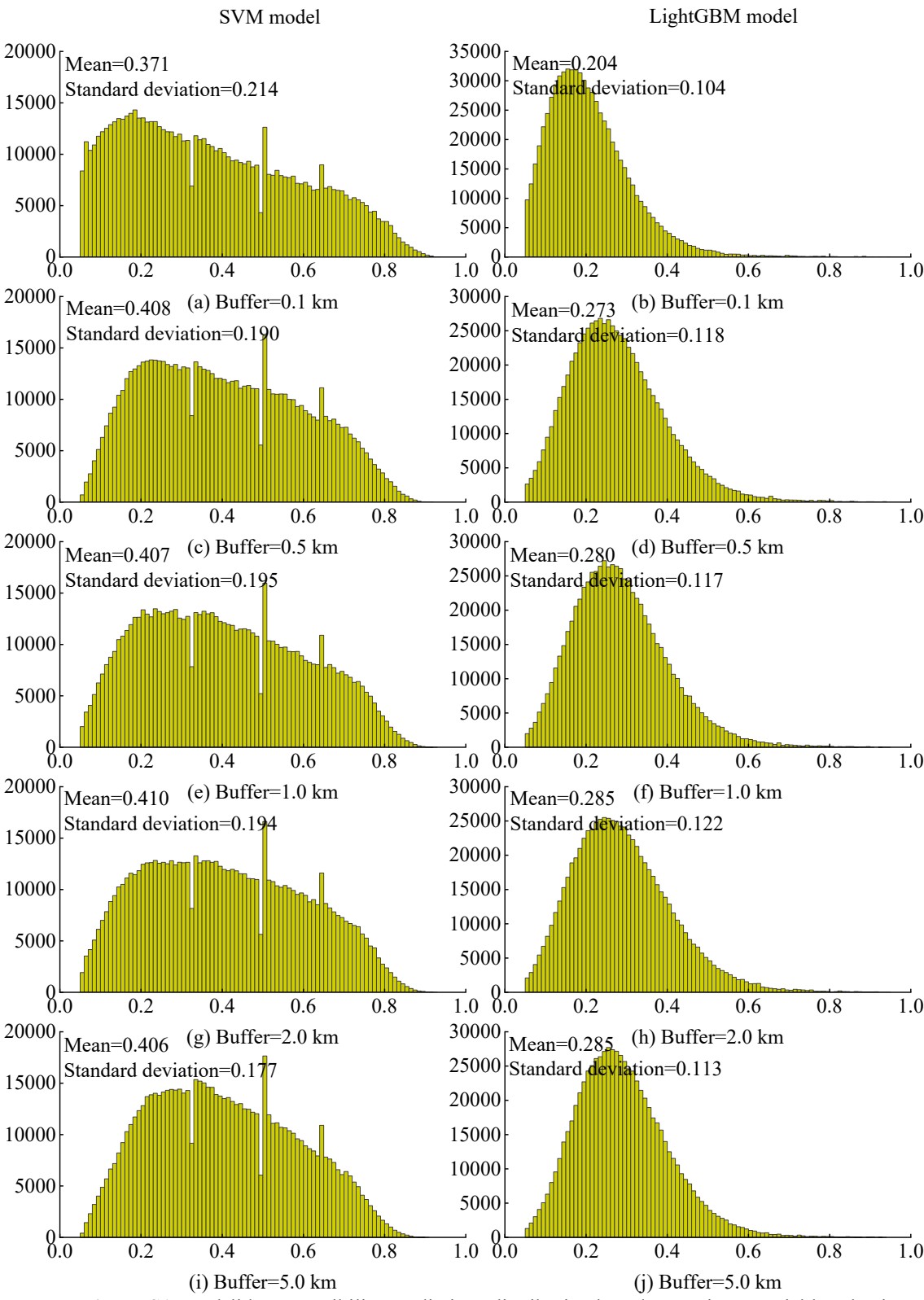

SVM model      LightGBM model

**Figure S1** Landslide susceptibility predictions distribution based on IV input variable selection.

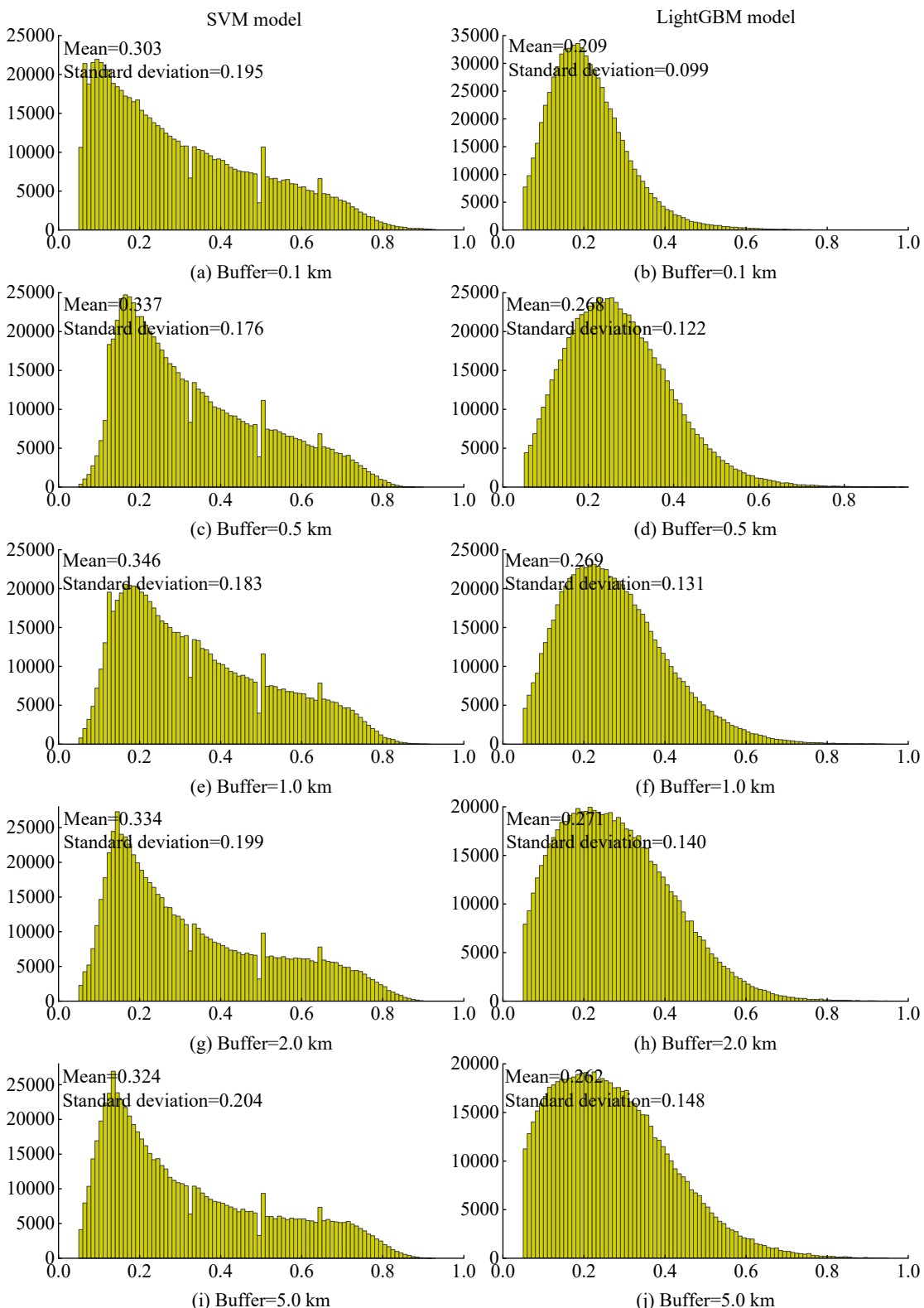

**Figure S2** Landslide susceptibility predictions distribution based on CF input variable selection.

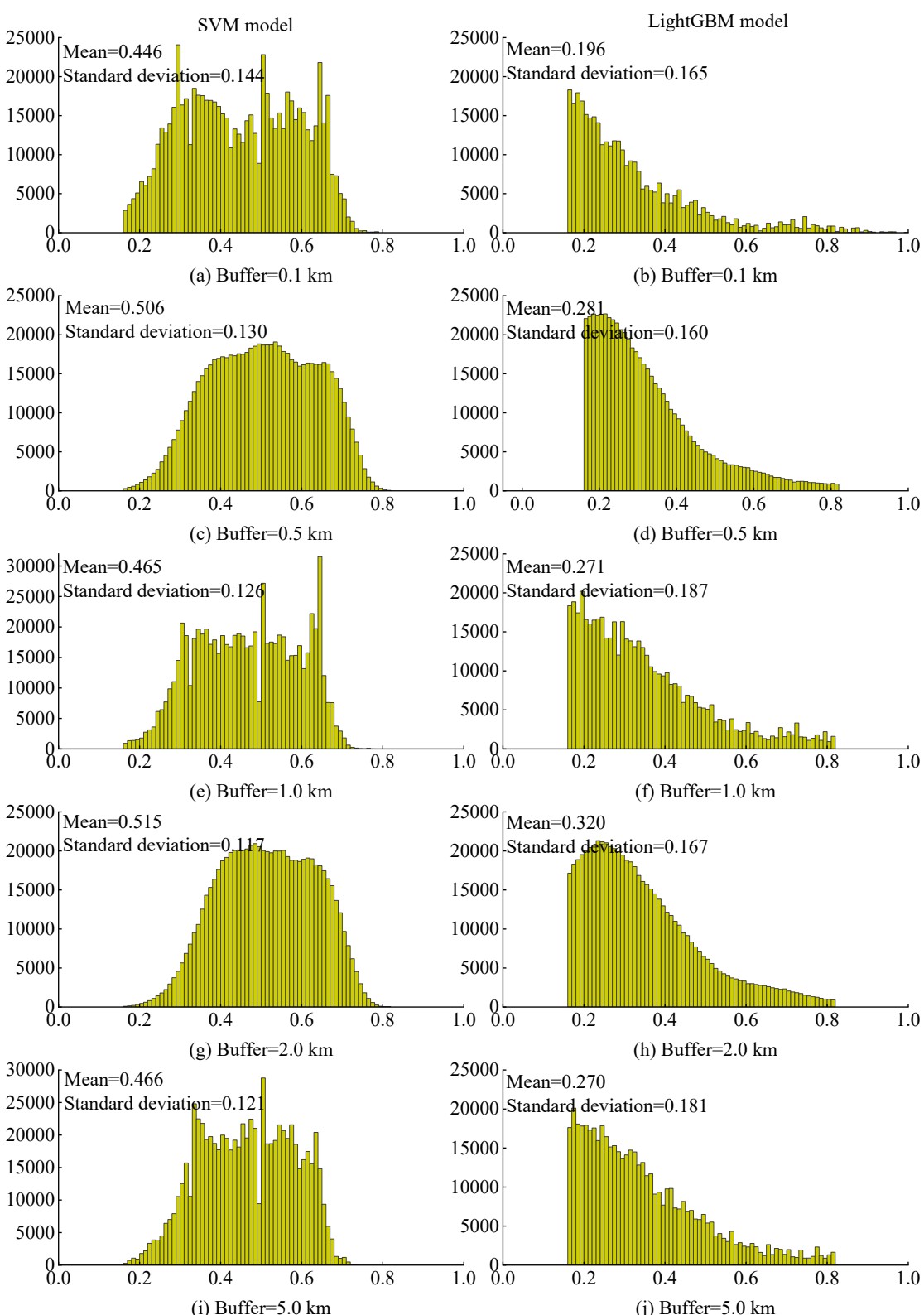

 **Figure S3** Landslide susceptibility predictions distribution based on FR input variable selection.