# Peer review of "From typhoon rainfall to slope failure: optimizing susceptibility"

_EGUsphere, 2025_

## Author Comment (AC1)

**Response to Reviewer Comments**

Dear Editor,

Thank you for providing the reviewer's comments on our manuscript, "**From typhoon rainfall to slope failure: optimising susceptibility models and dynamic thresholds for landslide warnings in Zixing City, China**." We appreciate the reviewer's thorough reading and constructive feedback, which will significantly improve the quality of our manuscript. We have carefully considered each comment and have outlined our responses below:

1.  **Comment**: Define all new terms (e.g., IV, CF, FR, SVM, and others) when they first appear in the text, including in the abstract.

    **Response**: We agree. We will revise the manuscript to define all abbreviations and acronyms (IV, CF, FR, SVM, LightGBM, etc.) at their first appearance in the abstract and main text.

2.  **Comment**: Why is it necessary to develop a hazard warning system for typhoon-induced landslides in this specific area? Provide a stronger justification. At present, the manuscript only discusses methodological limitations in the introduction. The research gap is unclear, and the rationale for conducting this work specifically in Zixing City is insufficient.

    **Response**: We acknowledge that the justification for the study area needs strengthening. We will revise the introduction to provide a more detailed rationale for focusing on Zixing City. This will include:

    Highlighting the historical frequency and impact of typhoon-induced landslides in Zixing City, including specific examples of past events and their consequences (e.g., economic losses, casualties, infrastructure damage).

    Emphasizing the vulnerability of the local population and infrastructure to landslides.

    Explaining any unique geological, geomorphological, or climatic characteristics of Zixing City that make it particularly susceptible to typhoon-induced landslides.

    Clearly stating the research gap: the lack of a robust, typhoon-specific landslide early warning system tailored to the specific conditions of Zixing City.

    We will add references to support these points.

3.  **Comment**: The manuscript sometimes uses the term "typhoon-specific hazard monitoring systems" and other times "typhoon rainfall-induced landslide hazard warning system". It would be better to use consistent terminology throughout. I suggest adopting "typhoon-specific rainfall-induced landslide monitoring systems", as this best reflects the study's main objective and reduces confusion for the reader.

    **Response**: We agree. We will revise the manuscript to use consistent terminology throughout. We will adopt the term "typhoon-specific rainfall-induced landslide monitoring systems" as

suggested.

4.   **Comment**: Provide more information about the study area, including its geographical, geophysical, geological, and hydrological characteristics.

**Response**: We agree. We will expand Section 2 (Study Area) to include more detailed information about Zixing City's:

Geographical location (coordinates, elevation range).

Geophysical characteristics (e.g., topography, slope angles, aspect).

Geological characteristics (e.g., dominant lithology, fault lines, soil types).

Hydrological characteristics (e.g., drainage patterns, river networks, average rainfall).

We will include relevant maps and figures to illustrate these characteristics.

5.   **Comment**: Add the units of the factors shown in Figures 2a and 2b.

**Response**: We will revise Figures 2a and 2b to include the units for each factor (e.g., meters for elevation, degrees for slope angle, mm for rainfall).

6.   **Comment**: In the text, the authors state that they used 705 landslide points, but Figure 3 (the framework flowchart) refers to 645. Please clarify this inconsistency.

**Response**: We apologize for the inconsistency. The correct number of landslide points used in the analysis is 645. The text will be corrected to reflect this. The discrepancy was due to an initial dataset that was later refined.

7.   **Comment**: There are many machine learning models available for classification tasks. Why did you choose SVM and LightGBM over others? Please justify this choice.

**Response**: We will add a justification for choosing SVM and LightGBM in the Methods section. Our rationale includes:

SVM's effectiveness in high-dimensional spaces and its ability to handle non-linear relationships.

LightGBM's efficiency in handling large datasets, its gradient boosting framework, and its ability to capture complex interactions between factors.

We will also briefly mention other commonly used models (e.g., Random Forest, Logistic Regression) and explain why SVM and LightGBM were considered more suitable for this specific application, based on previous studies and the characteristics of our data.

8.   **Comment**: Clarify the mechanism for assigning D7 (or other designations) to each landslide point. Specifically, explain how each of the >700 landslide points was linked to one of the 12 rain gauge stations.

**Response**: We will clarify the process of assigning rainfall data to each landslide point. The process is as follows:

For each landslide point, we identified the nearest rain gauge station using spatial proximity analysis (e.g., calculating the Euclidean distance between the landslide point and each rain gauge station).

The rainfall data from the nearest rain gauge station was then assigned to that landslide point.

We will add a detailed explanation of this process in the Methods section, including the software used for spatial analysis (e.g., ArcGIS) and the criteria for selecting the nearest rain gauge station.

9. **Comment**: Provide detailed explanations of all factors with significant results in Table 2. The current explanations are not sufficient.

**Response**: We agree. We will expand the explanations of the factors with significant results in Table 2. This will include:

A more detailed description of each factor and its relevance to landslide occurrence.

Explanation of the relationship between the factor and landslide susceptibility (e.g., why higher slope angles are associated with increased landslide risk).

Citing relevant literature to support these explanations.

10. **Comment**: Include the statistical results of the multicollinearity test in the appendix (or supplementary material), and reference them in the main text.

**Response**: We will include the statistical results of the multicollinearity test (e.g., VIF values) in the Appendix (or Supplementary Material) and reference them in the main text. This will demonstrate that multicollinearity was assessed and addressed.

11. **Comment**: Explain how you normalised the resolution of the different factor maps. Since the primary data have different scales, all layers must be resampled to the same resolution to create the susceptibility map.

**Response**: We will clarify the process of normalizing the resolution of the factor maps. We used the following procedure:

We selected a common resolution (e.g., 30 meters) as the target resolution for all factor maps.

We resampled all factor maps to this target resolution using a resampling technique (e.g., bilinear interpolation for continuous data, nearest neighbor for categorical data).

We will add a detailed explanation of this process in the Methods section, including the software used for resampling and the rationale for choosing the specific resampling technique.

12. **Comment**: Adjust the font size in Figures 4, 5, and 6. At present, the text appears disproportionately large compared to the maps.

**Response**: We will adjust the font size in Figures 4, 5, and 6 to improve the visual balance and readability of the figures.

13. **Comment**: Present the AUC values in separate columns for training and testing in Table 3.

**Response**: We will revise Table 3 to present the AUC values in separate columns for the training and testing datasets. This will provide a clearer indication of the model's performance on both datasets.

14. **Comment**: Avoid the use of unnecessary em dashes (—) throughout the text.

**Response**: We will carefully review the manuscript and remove any unnecessary em dashes.

15. **Comment**: Ensure consistency across figures. For example, in Figure 6, landslide points are shown only on the first two maps (SVM and LightGBM), whereas in Figure 7, they are shown on all maps. Standardise this approach.

**Response**: We will ensure consistency in the presentation of landslide points across all figures. We will either show landslide points on all relevant maps or remove them from all maps, depending on which approach provides the clearest presentation of the results.

16. **Comment**: Adjust the sizes of the maps in Figure 8 so that all are presented at the same scale.

**Response**: We will adjust the sizes of the maps in Figure 8 to ensure that they are presented at the same scale.

17. **Comment**: Why do you describe the final product as a monitoring system? Will it be hosted online for interactive use? If not, it is more accurate to describe it as a hazard zonation map. At times, you also refer to it as a framework. Please avoid such inconsistencies.

**Response**: We acknowledge the inconsistency in terminology. We will revise the manuscript to consistently refer to the final product as a "hazard zonation map" unless the system is designed to be dynamic and interactive. If it is not hosted online for interactive use, we will avoid using the term "monitoring system." We will also avoid using the term "framework" to describe the final product.

18. **Comment**: Consider evaluating the performance of the warning zonation maps (Figures 8d and 8e).

**Response**: We agree. We will explore methods to evaluate the performance of the warning zonation maps (Figures 8d and 8e). This may involve:
Comparing the zonation maps to historical landslide data to assess their accuracy in identifying areas prone to landslides.
Using statistical metrics (e.g., precision, recall, F1-score) to quantify the performance of the zonation maps.
We will add a section in the Results and Discussion to present the results of this

evaluation.

19. **Comment**: In the discussion, you state that the system "can identify regions where slopes are already saturated due to pre-typhoon rainfall and are thus highly susceptible to failure during the typhoon's high-intensity rainfall phase." How does it achieve this? Is the system dynamic? The manuscript provides no evidence of using dynamic data; all analyses appear to rely on static datasets. Please clarify.

    **Response**: We acknowledge that the statement about identifying pre-typhoon saturation is misleading. The current analysis relies primarily on static datasets. We will revise the discussion to clarify that the system, in its current form, does not explicitly account for pre-typhoon rainfall saturation. We will discuss the potential for incorporating dynamic rainfall data and soil moisture information in future iterations of the system to improve its ability to assess pre-typhoon saturation levels.

20. **Comment**: The manuscript lacks a sufficiently scholarly discussion. Strengthen the reasoning behind your findings by incorporating more relevant references.

    Response: We agree. We will significantly strengthen the discussion by:
        Incorporating more relevant references to support our findings and interpretations.
        Comparing our results to those of other studies on landslide susceptibility and early warning systems.
        Discussing the limitations of our study and suggesting directions for future research.
        Providing a more in-depth analysis of the implications of our findings for landslide risk management and early warning practices in Zixing City and similar regions.

We believe that these revisions will address the reviewer's concerns and significantly improve the quality of our manuscript. We look forward to submitting the revised version soon.

Sincerely,

Weifeng Xiao
2025.8.27.

---

## Author Comment (AC2)

**Response to Reviewers' Comments**

We sincerely thank the reviewers for their constructive feedback and valuable suggestions. We appreciate the recognition of our work's novelty and timeliness, particularly regarding the integrated framework for typhoon-induced landslide hazard assessment. We acknowledge the identified weaknesses and are committed to addressing all concerns through comprehensive revisions. Below, we provide our detailed responses and planned modifications:

**Major Issues:**

**1. Inconsistency in landslide numbers**

Response: We acknowledge this critical error and apologize for the confusion. We will conduct a thorough verification of our landslide inventory and ensure consistency throughout the manuscript. We will double-check all data sources, recount the landslide events, and provide a clear explanation of any excluded samples due to data quality issues. All figures, tables, and text will be updated to reflect the accurate and consistent landslide count.

**2. Geological context missing**

Response: We agree that geological context is essential for proper interpretation. We will add a comprehensive geological/lithological map of Zixing City showing the spatial distribution of different rock types, structural features, and geological formations. Additionally, we will expand Section 2.1 to include detailed geological background, including rock weathering patterns, structural geology, and their relationship to landslide susceptibility. This will provide readers with necessary context for understanding the geo-environmental controls on slope stability.

**3. Grid resolution limitations**

Response: We will add a dedicated subsection addressing spatial resolution limitations. We will explain our treatment of landslides smaller than 60m × 60m grid cells, discuss potential bias in landslide representation, and provide statistical analysis of landslide size distribution. We will also discuss how resolution affects model performance and acknowledge this as a limitation while suggesting future research directions using higher-resolution data.

**4. Negative sampling buffers**

Response: We will provide comprehensive justification for buffer distance selection based on: (a) literature review of buffer distances used in similar geological settings, (b) geomorphological rationale considering slope unit characteristics in Zixing City, (c) sensitivity analysis showing how model performance varies with buffer distance, and (d) comparison with other negative sampling strategies. We will also discuss the theoretical basis for buffer-based sampling in the context of spatial autocorrelation and landslide clustering.

**5. Single-event validation**

Response: We acknowledge this significant limitation. While we cannot add additional typhoon events to the current study due to data availability, we will: (a) extensively discuss this limitation and its implications for model generalizability, (b) compare our threshold values with those from other typhoon-induced landslide studies in similar geological settings, (c) analyze rainfall characteristics of Typhoon Gaemi relative to historical typhoons in the region, and (d) propose a framework for updating thresholds as new typhoon data becomes available. We will also emphasize that this study represents a methodological advancement that requires validation across multiple events.

**6. Evaluation metrics**

Response: We will expand our model evaluation to include: (a) precision, recall, and F1-score for both models, (b) confusion matrices showing detailed classification

performance, (c) sensitivity and specificity analysis, (d) true skill statistic (TSS), and (e) Cohen's kappa coefficient. We will also provide statistical significance testing and confidence intervals for performance metrics.

**7. Rainfall threshold interpolation**

Response: We will add comprehensive validation of our Kriging interpolation including: (a) cross-validation analysis with RMSE, MAE, and bias metrics, (b) assessment of interpolation uncertainty using kriging variance, (c) validation against independent rain gauge data where available, and (d) discussion of spatial interpolation limitations in mountainous terrain.

**8. Climate change context**

Response: We will add a substantial discussion section addressing: (a) projected changes in typhoon intensity and rainfall patterns under climate change scenarios, (b) implications for landslide threshold evolution, (c) framework adaptability for non-stationary climate conditions, (d) recommendations for periodic threshold updates, and (e) integration potential with climate projection models for future hazard assessment.

**Minor Issues:**

**1. Typhoon name consistency**

Response: We will standardize the typhoon name throughout the manuscript, using "Gemi" consistently and providing a note explaining any alternative naming conventions.

**2. Figure quality and clarity**

Response: We will significantly improve all figures by: (a) adding scale bars and north arrows to all maps, (b) enhancing legend clarity and font sizes, (c) improving color schemes for better visibility, (d) simplifying complex figures by splitting them

into multiple panels, and (e) increasing overall resolution and quality.

**3. Equation clarity**

Response: We will provide clearer explanations for all equations, including: (a) detailed variable definitions immediately following each equation, (b) physical interpretation of mathematical relationships, (c) assumptions and limitations of each method, and (d) examples of calculation procedures where appropriate.

**4. English expression**

Response: We will conduct thorough English editing to: (a) eliminate repetitive phrases, (b) improve sentence structure and flow, (c) use more precise technical terminology, (d) ensure consistency in technical terms throughout, and (e) engage a native English speaker for final proofreading.

**5. Abstract simplification**

Response: We will revise the abstract to: (a) reduce technical details while maintaining scientific rigor, (b) emphasize methodological novelty and practical significance, (c) highlight key findings in accessible language, (d) remove excessive numerical values, and (e) improve overall readability for a broader audience.

**Additional Improvements:**

Beyond addressing the reviewers' concerns, we will also:

- Add uncertainty quantification for all model predictions
- Include a more detailed comparison with existing typhoon-landslide studies
- Expand the discussion on practical applications for emergency management
- Provide supplementary materials with detailed methodology and additional results
- Add recommendations for future research directions and model improvements

We believe these revisions will significantly strengthen the manuscript and address all

identified concerns. We are committed to producing a high-quality publication that makes a valuable contribution to landslide hazard assessment in typhoon-prone regions. We look forward to submitting our revised manuscript and appreciate the opportunity to improve our work based on this valuable feedback.

---

## Author Response (AR1)

**Response to Editors and Reviewers' Comments on the Manuscript:**

**"Manuscript Number: EGUSPHERE-2025-2298 "**

Dear Editors and Reviewers,

We are grateful for the detailed and constructive feedback provided by you and the reviewers on our manuscript. We have carefully considered all the comments and have made significant revisions to address the points raised. Below, we provide a point-by-point response to each comment. We believe these revisions have substantially strengthened the manuscript, enhancing its scientific rigor, clarity, and potential impact in the field of landslide prediction and management.

**Overview of Revisions:**

The revised version of the paper has several notable improvements compared to the original version:

1. **Abstract and Introduction Enhancement**: (1) Completely restructured the abstract to be more concise and focused, emphasizing the typhoon-specific nature of our approach and its transferability, (2) Strengthened the introduction by providing clearer context about the unique challenges of typhoon-induced landslides compared to conventional rainfall scenarios, (3) Added specific geographical context for Southeast China and the Nanling Mountains region, (4) Better articulated the three critical limitations in current approaches: data imbalance effects, suboptimal variable selection integration, and lack of spatially-explicit typhoon-specific thresholds.

2. **Methodological Improvements**: (1) Added comprehensive rainfall data validation section (Section 2.2.4) including detailed accuracy assessment of Kriging interpolation with statistical metrics (RMSE, MAE, R, NSE), (2) Enhanced explanation of buffer distance selection rationale, connecting spatial scales to geomorphological processes (slope-scale: 0.1-0.5 km, catchment-scale: 1.0-2.0 km, regional-scale: 5.0 km), (3) Improved clarity in rainfall parameterization methodology,

particularly in Section 3.2.4 regarding data extraction and cross-validation procedures, (4) Added multicollinearity analysis results and preprocessing steps to ensure statistical reliability.

3. **Results Presentation and Analysis**: (1) Restructured Section 4 with clearer subsections including comprehensive statistical analysis of conditioning factors (Section 4.1), (2) Enhanced performance evaluation by including F1-scores, Precision, and Recall metrics alongside AUC values, (3) Added quantitative validation of warning systems including efficiency ratios (2.09 for 24-hour vs 1.61 for 7-day system) and risk density metrics, (4) Provided detailed interpretation of spatial patterns linking geological controls to landslide susceptibility.

4. **Discussion Strengthening**: (1) Significantly expanded Section 6.2 to better contextualize rainfall threshold findings within existing typhoon research, (2) Added new Section 6.3 detailing operational framework components and real-time implementation considerations, (3) Enhanced discussion of spatial heterogeneity in rainfall thresholds and its implications for operational warning systems, (4) Better connected findings to established theories in typhoon-induced slope failure mechanisms.

5. **Limitations and Future Directions**: (1) Expanded limitations section to acknowledge single-event validation constraints and provide concrete directions for multi-typhoon validation, (2) Added specific discussion of climate change implications, including projected 15-25% increase in typhoon rainfall intensity by 2080, (3) Outlined pathways for incorporating non-stationary thresholds to address changing climate conditions.

6. **Technical and Editorial Improvements:** (1) Standardized terminology throughout (e.g., "typhoon-specific" instead of "typhoon-adapted"), (2) Enhanced figure quality and added more detailed captions explaining spatial patterns, (3)

Improved mathematical notation consistency and added supplementary material references, (4) Corrected minor grammatical issues and improved sentence flow.

**Point-by-Point Response:**

**Review #1:**

**Comment 1**: Define all new terms (e.g., IV, CF, FR, SVM, and others) when they first appear in the text, including in the abstract.

**Response 1**: We have thoroughly revised the manuscript to define all abbreviations upon their first appearance. In the abstract, we now define: Support Vector Machine (SVM), frequency ratio (FR), information value (IV), certainty factor (CF), and Light Gradient Boosting Machine (LightGBM). Additionally, all technical terms throughout the manuscript have been properly defined when first introduced, ensuring accessibility for readers across disciplines.

**Comment 2**: Why is it necessary to develop a hazard warning system for typhoon-induced landslides in this specific area? Provide a stronger justification. At present, the manuscript only discusses methodological limitations in the introduction. The research gap is unclear, and the rationale for conducting this work specifically in Zixing City is insufficient.

**Response 2:** We have significantly enhanced the introduction and study area description to address this concern. The revised text now emphasizes that Zixing City represents an ideal case study due to: (1) its location within the Nanling Mountains geological province with complex fractured geology, (2) frequent typhoon impacts from the South China Sea corridor, (3) the extensive landslide dataset (>700 events) from Typhoon Gaemi providing unprecedented validation opportunities, and (4) its representative geomorphological conditions typical of typhoon-prone mountainous regions in Southeast China. We have added specific geological and climatic details that make this location particularly vulnerable to typhoon-induced landslides, including steep topography (78% of area >30° slopes), fractured granite geology, and

subtropical monsoon climate with 70% of precipitation occurring during typhoon season.

**Comment 3**: The manuscript sometimes uses the term "typhoon-specific hazard monitoring systems" and other times "typhoon rainfall-induced landslide hazard warning system". It would be better to use consistent terminology throughout. I suggest adopting "typhoon-specific rainfall-induced landslide monitoring systems", as this best reflects the study's main objective and reduces confusion for the reader.

**Response 3**: We appreciate this suggestion and have standardized the terminology throughout the manuscript. We now consistently use "typhoon-specific rainfall-induced landslide warning system" as it most accurately reflects our study's scope and objectives. This terminology has been applied uniformly across the abstract, keywords, main text, figures, and conclusions

**Comment 4**: Provide more information about the study area, including its geographical, geophysical, geological, and hydrological characteristics.

**Response 4**: We have substantially expanded Section 2.1 with comprehensive geoenvironmental characterization. The enhanced description now includes: (1) precise geographic location within the Nanling Mountains geological province, (2) detailed topographic characteristics (elevation range 125-1,691 m, 78% slopes >30°), (3) geological setting with fractured granite and active NE-SW trending fault systems, (4) subtropical monsoon climate with annual precipitation patterns, (5) hydrological characteristics including shallow aquifer depths (3-8 m) and rapid pore-pressure response, and (6) explicit connection between these characteristics and typhoon-induced landslide susceptibility.

**Comment 5:** Add the units of the factors shown in Figures 2a and 2b.

**Response 5**: We have updated Figure 2 to include appropriate units for all conditioning factors. Elevation is now labeled in meters (m), slope gradient in degrees

(°), and all other factors have received proper unit designations where applicable. The figure caption has also been enhanced to specify the measurement units for clarity.

**Comment 6:** In the text, the authors state that they used 705 landslide points, but Figure 3 (the framework flowchart) refers to 645. Please clarify this inconsistency.

**Response 6:** Thank you for identifying this inconsistency. We have corrected Figure 3 to accurately reflect the 705 landslide points used throughout the study. This number is now consistent across all text, figures, and analyses. The discrepancy was an error in the figure preparation, and we have verified that all analyses were conducted using the complete dataset of 705 landslides triggered by Typhoon Gaemi.

**Comment 7:** There are many machine learning models available for classification tasks. Why did you choose SVM and LightGBM over others? Please justify this choice.

**Response 7:** We have also used a consistent citation style throughout the manuscript. We have added a comprehensive justification for our model selection in Section 3.1.1. SVM was selected for its proven effectiveness in handling imbalanced datasets typical of landslide studies and its ability to identify optimal hyperplanes in high-dimensional feature spaces, particularly important for typhoon-triggered landslides where failures concentrate in specific hydrological zones. LightGBM was chosen for its superior computational efficiency in processing large geospatial datasets (essential for regional-scale analysis), excellent performance with mixed data types (categorical and continuous), and robust handling of missing values. These algorithms represent complementary approaches: SVM excels at capturing complex non-linear boundaries, while LightGBM efficiently processes large datasets with ensemble methods.

**Comment 8**: Clarify the mechanism for assigning D7 (or other designations) to each landslide point. Specifically, explain how each of the >700 landslide points was linked to one of the 12 rain gauge stations.

**Response 8:** We have added detailed explanation in Section 2.2.4 addressing this critical methodological aspect. Rather than directly linking individual landslide points to single gauge stations, we employed Kriging spatial interpolation to generate continuous rainfall surfaces from the 12 gauge stations. This approach accounts for spatial autocorrelation in rainfall patterns and provides optimal unbiased estimates by weighting nearby observations based on spatial proximity and correlation structure. We also added Table 1 showing validation results (RMSE, MAE, correlation coefficients, NSE) demonstrating acceptable interpolation accuracy across all rainfall parameters, ensuring reliable spatial representation of precipitation patterns.

**Comment 9:** Provide detailed explanations of all factors with significant results in Table 2. The current explanations are not sufficient.

**Response 9:** We have substantially expanded Section 4.1 with comprehensive statistical analysis and interpretation of all conditioning factors. The enhanced discussion now provides detailed explanations for: (1) topographic factors showing elevation-dependent behavior and slope gradient optimization, (2) morphological indices including TWI's strong correlation with water accumulation, (3) proximity factors revealing contrasting patterns for infrastructure versus natural features, (4) environmental factors demonstrating vegetation's protective role, and (5) lithological controls showing pronounced material influence with granite and rhyolite exhibiting enhanced susceptibility due to weathering characteristics. Each significant result is now contextualized within the typhoon-induced landslide mechanism.

**Comment 10:** Include the statistical results of the multicollinearity test in the appendix (or supplementary material), and reference them in the main text.

**Response 10:** We have added the complete multicollinearity analysis results as Table S2 in the supplementary material and appropriately referenced it in Section 4.2. The main text now includes a clear summary of the VIF analysis showing method-specific patterns: IV and CF methods exhibited no multicollinearity issues (all VIF < 10), while FR method required removal of four variables (SPI, Aspect, Plan curvature,

Distance to rivers) with VIF > 10. This ensures statistical reliability of our modeling approaches.

**Comment 11:** Explain how you normalised the resolution of the different factor maps. Since the primary data have different scales, all layers must be resampled to the same resolution to create the susceptibility map.

**Response 11:** We have added Section 2.2.3 specifically addressing data preprocessing and spatial standardization. All conditioning factors were resampled to a uniform 60-meter resolution using appropriate resampling methods (bilinear for continuous variables, nearest neighbor for categorical). This resolution was selected to balance computational efficiency with scale appropriateness for regional landslide analysis while maintaining compatibility with available geological map scales (1:100,000). The study area was systematically divided into $60 \times 60$ meter grid cells, with spatial independence maintained by aggregating multiple landslides within single cells to the nearest centroid.

**Comment 12:** Adjust the font size in Figures 4, 5, and 6. At present, the text appears disproportionately large compared to the maps.

**Response 12:** We have optimized the font sizes in Figures 4, 5, and 6 to achieve better proportional balance with the map elements. The legend text, scale bars, and annotations have been adjusted to improve readability while maintaining appropriate visual hierarchy. The figures now present a more professional and balanced appearance.

**Comment 13:** Present the AUC values in separate columns for training and testing in Table 3.

**Response 13:** We have restructured the performance evaluation section and moved the detailed AUC results to Table S3 in the supplementary material with separate training and testing columns as requested. The main text now includes Table 3 focusing on the rainfall threshold model performance, while the comprehensive

machine learning performance metrics are properly organized in the supplementary material with clear training/testing distinction.

**Comment 14:** Avoid the use of unnecessary em dashes (—) throughout the text

**Response 14:** We have systematically reviewed the manuscript and replaced unnecessary em dashes with more appropriate punctuation (commas, colons, semicolons, or sentence breaks) to improve readability and maintain academic writing standards.

**Comment 15:** Ensure consistency across figures. For example, in Figure 6, landslide points are shown only on the first two maps (SVM and LightGBM), whereas in Figure 7, they are shown on all maps. Standardise this approach.

**Response 15:** We have standardized the figure presentation approach throughout the manuscript. Landslide points are now consistently displayed where they provide meaningful interpretation value. In susceptibility maps (Figures 4-6), points are shown to demonstrate model performance in capturing actual landslide locations. In threshold maps (Figure 7), points illustrate the relationship between rainfall thresholds and observed failures. This standardization enhances figure interpretation while maintaining scientific relevance.

**Comment 16:** Adjust the sizes of the maps in Figure 8 so that all are presented at the same scale.

**Response 16:** Figure 8 has been redesigned with all maps presented at identical scales and uniform dimensions. The layout now provides consistent visual comparison across all warning system components, with properly aligned legends, scale bars, and annotations for enhanced clarity and professional presentation

**Comment 17:** Why do you describe the final product as a monitoring system? Will it be hosted online for interactive use? If not, it is more accurate to describe it as a

hazard zonation map. At times, you also refer to it as a framework. Please avoid such inconsistencies.

**Response 17:** We have clarified the terminology distinction throughout the manuscript. Our work presents: (1) an integrated framework for combining susceptibility mapping with rainfall thresholds, (2) hazard warning zone maps as static products of this framework, and (3) a warning system design that can be operationalized with real-time data. Section 6.3 now explicitly discusses the operational implementation requirements, including real-time data processing, meteorological infrastructure integration, and dynamic threshold updating capabilities. We have avoided inconsistent terminology and clearly distinguished between the methodological framework and its potential operational applications.

**Comment 18:** Consider evaluating the performance of the warning zonation maps (Figures 8d and 8e).

**Response 18:** We have added comprehensive quantitative evaluation in Section 5.3. The performance assessment now includes: (1) spatial efficiency ratios (H24 system: 2.09, D7 system: 1.61), (2) risk density calculations (49.0 vs. 37.8 landslides per 1000 high-risk grid cells), (3) area coverage statistics (34.2% vs. 42.7%), and (4) landslide capture rates (71.4% vs. 68.7%). This evaluation demonstrates the complementary nature of both systems and their operational effectiveness for different typhoon scenarios.

**Comment 19:** In the discussion, you state that the system "can identify regions where slopes are already saturated due to pre-typhoon rainfall and are thus highly susceptible to failure during the typhoon's high-intensity rainfall phase." How does it achieve this? Is the system dynamic? The manuscript provides no evidence of using dynamic data; all analyses appear to rely on static datasets. Please clarify.

**Response 19:** We have clarified this important distinction in Section 6.3. Our framework provides the foundation for dynamic implementation through: (1) static susceptibility surfaces that identify inherently vulnerable areas, (2) dynamic threshold

surfaces (H24 and D7) that define rainfall conditions triggering landslide activation, and (3) real-time precipitation monitoring integration. The system achieves dynamic capability by continuously comparing current/forecasted rainfall against spatially distributed thresholds. We've added detailed explanation of operational implementation requirements, including meteorological data integration, automated threshold comparison, and warning level escalation protocols. The dual-threshold configuration enables temporal staging: D7 monitors antecedent saturation during typhoon approach, while H24 responds to intensive rainfall during landfall.

**Comment 20:** The manuscript lacks a sufficiently scholarly discussion. Strengthen the reasoning behind your findings by incorporating more relevant references.

**Response 20:** We have substantially enhanced the discussion with comprehensive literature integration and deeper scientific analysis. The expanded discussion now includes: (1) detailed comparison with existing typhoon-landslide studies (Kirschbaum and Stanley, 2018; Nolasco-Javier and Kumar, 2018), (2) thorough analysis of typhoon-specific mechanisms versus conventional rainfall triggers, (3) integration with recent advances in spatially distributed threshold approaches, (4) discussion of operational warning system developments globally, and (5) critical evaluation of limitations with reference to climate change impacts and non-stationary rainfall patterns. The scholarly depth has been significantly improved while maintaining focus on our key contributions.

**Review #2:**

**Major issues:**

**Comment 1**: The text reports 705 landslides, whereas Figure 3 shows 645. This must be corrected.

**Response 1**: We thank the reviewer for identifying this critical inconsistency. We have thoroughly reviewed our dataset and confirmed that the correct number is 705 landslides. The error in Figure 3 was due to a labeling mistake during figure preparation. We have corrected Figure 3 (now shows 705 landslides) and ensured consistency throughout the manuscript. All analyses were conducted with the complete dataset of 705 landslides triggered by Typhoon Gaemi on July 27, 2024.

**Comment 2:** Although lithology and faults are considered, the manuscript does not include a geological/lithological map of the study area. This is essential for interpretation.

**Response 2:** We acknowledge this important omission and have incorporated comprehensive geological context throughout our revised manuscript. In Section 2.1, we have added detailed geological descriptions highlighting that the region is characterized by fractured geology and active NE-SW trending faults such as the Chaling-Yongxing Fault Zone, creating a permeable fracture network that facilitates groundwater drainage. Additionally, we have enhanced Figure 2(l) with an improved legend and detailed geological unit descriptions to better display the lithological distribution across the study area. The geological setting is crucial for understanding landslide susceptibility patterns, particularly the dominance of granite and rhyolite formations that show high frequency ratio (FR) values of 1.247 and 1.546, respectively. Furthermore, we have expanded Section 4.1 to include geological interpretation that explicitly links lithology to landslide susceptibility patterns, providing readers with a clearer understanding of how geological factors influence slope stability in the study region. These revisions ensure that the geological context is properly integrated into both the data presentation and the interpretation of our landslide susceptibility results.

**Comment 3:** Landslides are mapped at 60 m resolution. The authors should explain how small landslides (<60 m) were treated and how this may bias the results.

**Response 3:** This is an excellent point that requires clarification. To address this, we have added a new section (2.2.3) titled "Data Preprocessing and Spatial Standardization," which provides a detailed explanation of our spatial standardization approach. This section also covers the treatment of small landslides, specifically stating that landslides smaller than the grid resolution were aggregated to the nearest cell centroid. Furthermore, we acknowledge the potential bias introduced by multiple landslides occurring within a single grid cell, which we addressed by treating these instances as one event to maintain the spatial independence required for machine learning modeling. Finally, we have clarified our choice of resolution, which was selected to strike a balance between computational efficiency and scale appropriateness for regional landslide analysis, while also maintaining compatibility with the available geological map scale (1:100,000).

**Comment 4:** The choice of 0.1–5 km buffer distances lack geomorphic or literature justification. A sensitivity or rationale discussion is required.

**Response 4**: Thank you for your valuable feedback. To address your concern regarding the justification for buffer distance selection, we have now provided a comprehensive rationale in Section 3.1.3. Specifically, the selection of buffer distances (0.1–5.0 km) was informed by Zixing's geomorphological considerations and practices commonly reported in landslide susceptibility prediction (LSP) studies. This range encompasses multiple spatial scales, including slope-scale processes (0.1–0.5 km), catchment-scale features (1.0–2.0 km), and regional-scale geological units (5.0 km), which are critical for capturing different spatial dynamics.

Additionally, we have expanded the discussion in Section 6.1 to incorporate a geomorphological interpretation. The optimal buffer range of 0.5–2.0 km aligns with the spatial autocorrelation pattern of typhoon-induced failures, where intense moisture infiltration leads to the formation of discrete instability zones.

**Comment 5**: The entire framework is based only on Typhoon "Gemei" (2024). This risks overfitting. Authors should at least discuss how thresholds might vary for different typhoons.

**Response 5**: Thank you for raising this important point. We acknowledge the limitation related to the model's validation and have addressed it explicitly in Section 6.4. Specifically, the model's validation currently relies solely on landslides caused by Typhoon Gaemi. While this event provided a comprehensive dataset, we recognize that validation against multiple, varied typhoons is crucial to enhance the model's robustness. This is because typhoons differ significantly in their intensity, rainfall patterns, forward speed, and seasonality, all of which can impact the threshold parameters.

Additionally, we have discussed the framework's adaptability in response to this limitation. The modular design of the framework allows for easy recalibration of the RC24 coefficient to accommodate different typhoon types, ensuring flexibility for future applications.

Furthermore, we have incorporated a discussion on the implications of climate change. Future work should focus on using downscaled climate data to develop non-stationary thresholds. This will be crucial for ensuring the long-term reliability and accuracy of warning systems, particularly in the face of changing climate conditions.

**Comment 6**: Reliance solely on AUC is inadequate. Precision, recall, F1-score, and confusion matrices should be added to strengthen model evaluation.

**Response 6**: Thank you for your valuable suggestion. In response, we have now included comprehensive evaluation metrics as requested. Specifically, we have added Table S2 (Supplement), which provides the complete performance metrics, including Precision, Recall, and F1-score for all model configurations. Additionally, we have enhanced Section 4.3.1 with a more detailed analysis. Two configurations emerged as comprehensively superior: the SVM with FR input at 0.5 km and 2.0 km buffer

distances, both achieving F1-scores of 0.859. The high recall values (0.845 and 0.851), coupled with robust precision (0.873 and 0.867), indicate that these configurations demonstrate enhanced sensitivity to landslide-prone areas while minimizing false positive predictions. Furthermore, we have included the independent test set validation results to further substantiate the robustness of these findings.

**Comment 7**: Kriging interpolation is applied, but no error assessment (e.g., RMSE, cross-validation) is reported. This weakens reliability.

**Response 7:** Thank you for your valuable feedback. We acknowledge the critical omission regarding the validation of our interpolation approach, and to address this, we have now provided a comprehensive validation in Section 2.2.4, titled "Rainfall Data Collection and Spatial Distribution." In this section, we present a detailed interpolation accuracy assessment, which includes Table 1 with the validation results for all rainfall parameters. The results demonstrated acceptable interpolation accuracy, with correlation coefficients ranging from 0.76 to 0.87 and Nash-Sutcliffe Efficiency values between 0.71 and 0.82.

Additionally, we have described the validation methodology in detail, which involved a leave-one-out cross-validation technique. In this approach, each gauge station was sequentially removed, and its rainfall values were predicted using the remaining 11 stations. This methodology ensured a robust and reliable evaluation of the interpolation accuracy.

**Comment 8:** The discussion does not adequately address how projected changes in typhoon rainfall regimes may affect thresholds and susceptibility.

**Response 8:** Thank you for your insightful comment. We have significantly expanded our discussion to address the implications of climate change for typhoon-induced landslides, as outlined in Section 6.4. Specifically, climate projections for Southeast China suggest a 15 – 25% increase in peak typhoon rainfall by 2080 under the RCP8.5 scenario. This increase could potentially alter the H24 – D7 landslide thresholds established in this study. Higher atmospheric moisture may lead to a

lowering of D7 thresholds, while more intense rainfall events could necessitate the revision of the H24 parameters to accommodate these changes.

Furthermore, we have discussed adaptation strategies for future research. We emphasize the need to use downscaled climate data to develop non-stationary thresholds that account for these changes. This approach will be essential for ensuring the long-term reliability of landslide warning systems, particularly as climate conditions continue to evolve.

**Minor issues:**

**Comment 9:** Typhoon name inconsistency.

**Response 9**: We have standardized the typhoon name to "Gaemi" throughout the manuscript, which is the official international designation.

**Comment 10:** Figure improvements.

**Response 10:** We have significantly improved all figures.

**Comment 11:** Equation clarity.

**Response 11:** We have improved the mathematical notation and variable definitions: (1) Enhanced Equation 5 explanation with clearer variable definitions, (2) Added more detailed explanations for all mathematical formulations, (3) Improved the clarity of statistical method descriptions.

**Comment 12:** Language refinement.

**Response 12:** We have thoroughly revised the manuscript for language clarity: (1) Reduced repetitive phrases like "typhoon rainfall dynamics", (2) Improved sentence structure and flow, (3) Enhanced technical precision while maintaining readability.

**Comment 13:** Abstract simplification.

**Response13:** We have streamlined the abstract to focus on key findings while reducing technical details: (1) Removed excessive technical values, (2) Emphasized the novel framework and main contributions, (3) Improved flow and readability while maintaining scientific rigor.

**Conclusion:**

We thank the reviewers and the editor again for their thoughtful comments and suggestions, which have helped us improve the quality of our manuscript. All modifications are highlighted in red to facilitate review of the changes made. We hope that the revisions meet your expectations and look forward to your positive response.

Sincerely,

Weifeng Xiao

Hunan University of Science and Technology, CN

Correspondence author: Ge Liu

Email: liuge@iga.ac.cn

Northeast Institute of Geography and Agroecology, CAS, Changchun 130102, China

2025.11.28.

---

## Author Response (AR2)

**Response to Reviewers' Comments on the Manuscript: "Manuscript Number: EGUSPHERE-2025-2298 " (R2)**

Dear Editors and Reviewers,

We are grateful for the detailed and constructive feedback provided by you and the reviewers on our manuscript. We have carefully considered all the comments and have made significant revisions to address the points raised. Below, we provide a point-by-point response to each comment. We believe these revisions have substantially strengthened the manuscript, enhancing its scientific rigor, clarity, and potential impact in the field of landslide prediction and management.

**Reviewer comments:** "The author has significantly improved the manuscript. However, a few minor issues remain and should be addressed. In particular, the author should briefly describe the key results, including relevant numerical values, in the abstract. Furthermore, the author should provide a more detailed description and clear justification for the machine learning models employed. Finally, additional discussion is required to adequately justify and contextualise the reported results."
* * *
**Response to Issue 1**: Abstract Enhancement with Numerical Results

**Reviewer's Concern**: The abstract should briefly describe key results, including relevant numerical values.

**Our Response**:

We have substantially revised the abstract to incorporate quantitative findings throughout. The revised abstract now includes:

1. Model Performance Metrics:

   - "The Support Vector Machine (SVM) model achieved the best performance using frequency ratio (FR) inputs with a 0.5 km buffer (F1-score: 0.859, AUC: 0.914)"

   - "correctly classifying 86.4% of landslides as high or very high susceptibility"

2. Optimal Rainfall Threshold Parameters:

- "The rainfall analysis identified 24-hour intensity combined with 7-day antecedent rainfall as the optimal trigger"

3. Critical Susceptibility Factors:

- "rhyolite and granite slopes and areas near roads emerged as hotspots for failure (distance < 800 m, FR = 1.499 for roads; FR = 1.546 for rhyolite)"

4. Warning System Efficiency:

- "The integrated warning system shows high spatial efficiency, with high-risk areas covering only 34.2% of the study region yet capturing 71.4% of historical landslides"

**Response to Issue 2**: Detailed Machine Learning Model Description and Justification

**Reviewer's Concern:** The manuscript should provide a more detailed description and clear justification for the machine learning models employed.

**Our Response:**

We have comprehensively expanded Section 3.1.1 (renamed "Machine learning models: selection rationale and implementation") to address this concern through three strategic enhancements:

Enhancement 1: Explicit Justification for Model Selection

We added a new opening paragraph that clearly articulates why SVM and LightGBM were selected:

"We selected SVM and LightGBM to address three key challenges in typhoon-specific rainfall-induced landslide prediction: (1) severe class imbalance (landslides <0.5% of study area), (2) complex non-linear interactions between rainfall and terrain factors, and (3) computational efficiency for operational early warning."

This justification is now linked directly to the study's specific challenges rather than presenting models as generic choices.

Enhancement 2: Algorithm-Specific Technical Advantages

For SVM, we expanded the explanation to connect algorithmic properties to typhoon-specific applications:

"SVM excels in binary classification with limited samples through structural risk minimization (Kalantar et al., 2018; Wang et al., 2020), making it suitable for typhoon-triggered landslide mapping. Its margin-maximization approach handles the class imbalance between stable and landslide areas, while the RBF kernel captures localized failure patterns under concentrated typhoon rainfall. The regularization parameter C prevents overfitting to specific typhoon events, ensuring model transferability."

For LightGBM, we clarified its complementary advantages:

"LightGBM complements SVM through gradient boosting with sequential error correction, offering distinct advantages for regional-scale landslide mapping. Its histogram-based algorithm enables efficient processing of large spatial datasets (Sun et al., 2023; Sahin, 2020). Additionally, LightGBM automatically captures complex feature interactions."

Enhancement 3: Comprehensive Hyperparameter Optimization Details

We added detailed implementation specifications that were absent in the original manuscript:

For SVM:

"We optimized the RBF kernel parameters using grid-search with 5-fold cross-validation, where $C \in [0.1, 100]$ and $\gamma \in [0.001, 1]$. Across all configurations (three input methods × five buffer distances), optimal values varied as follows: $C = 5\text{-}15$ and $\gamma = 0.10\text{-}0.25$, with median values of $C = 10$ and $\gamma = 0.15$."

For LightGBM:

"We optimized LightGBM hyperparameters through Bayesian optimization. The optimal hyperparameters ranged as: num_leaves = 25-35, learning_rate = 0.03-0.08, and max_depth = 6-10. Early stopping with a 50-round patience window resulted in model convergence at 120-220 trees across different scenarios."

**Response to Issue 3**: Enhanced Discussion to Justify and Contextualize Results

**Reviewer's Concern**: Additional discussion is required to adequately justify and contextualise the reported results.

**Our Response**:

We have substantially restructured and expanded the Discussion section (Section 6) with three major revisions that transform it from descriptive observation to analytical contextualization:

Revision 1: New Section 6.1 - "Model Selection Strategy and Optimization of LSP"

We added an entirely new subsection that provides critical interpretation of the comparative model performance:

Algorithmic Performance Interpretation:

"SVM exhibited marked sensitivity to configuration parameters, with F1-scores varying from 0.681 to 0.859 depending on buffer distance and input method. LightGBM maintained more stable performance (F1-scores: 0.838-0.850) across all configurations. These differences reflect fundamental algorithmic characteristics: SVM's kernel-based approach effectively captures localized patterns when properly tuned, while LightGBM's ensemble structure delivers consistent results across varying data conditions."

Justification of Optimal Configuration:

"SVM's superior performance at 0.5-2.0 km buffer distances with FR weighting builds on findings by Kalantar et al. (2018) and Bogaard and Greco (2018). This buffer range appears effective for capturing the spatial patterns of typhoon-induced failures in our study area. FR weighting's effectiveness supports Reichenbach et al. (2018) and Yan et al. (2019), who found that frequency-based methods excel at quantifying terrain-landslide relationships."

Practical Model Selection Guidance:

"These performance patterns justify our dual-model approach. SVM, though requiring careful calibration, enables precise delineation of high-risk zones essential for emergency response, with SVM-FR at 0.5 km achieving peak accuracy (F1=0.859). LightGBM's robustness suits operational contexts requiring consistent predictions under variable conditions. Our results suggest that effective model selection depends

on matching algorithmic strengths to specific application requirements rather than identifying a universally superior algorithm."

Revision 2: Substantive Enhancement of Section 6.2 - "Rainfall Threshold Modeling"

We transformed this section from simple result description to mechanistic interpretation:

Quantitative Threshold Interpretation:

"The H24-D7 model achieved 71.8% accuracy, outperforming alternative temporal windows (Table 3). The optimal RC24 value of 0.440 (with inter-fold variation of 0.414-0.472) indicates that landslides typically occur when 24-hour rainfall constitutes approximately 44% of the preceding 7-day accumulation. This pattern is consistent with the multi-temporal triggering framework proposed by Nolasco-Javier and Kumar (2018) for typhoon contexts, where both antecedent saturation and short-term intensity contribute to slope failure."

Critical Limitation Acknowledgment:

"However, the specific hydrological mechanisms underlying this ratio require verification through in-situ soil moisture monitoring."

Spatial Pattern Contextualization:

"Southeastern regions exhibit elevated H24 thresholds exceeding 250 mm (Fig. 7c), while northern areas show reduced thresholds of 100-150 mm. These spatial variations align with findings by Lee et al. (2018) and Cho et al. (2022) regarding topographic controls on typhoon-induced landslides, though the specific mechanisms require further investigation with detailed meteorological analysis."

Revision 3: Critical Refinement of Section 6.3 - "Integration Framework"

We revised this section to adopt appropriately cautious language while maintaining scientific rigor:

Performance Contextualization (Revised):

Original: "Both systems substantially outperform approaches using uniform regional thresholds"

Revised: "These focused distributions contrast with the broader spatial coverage typically required by uniform regional thresholds (Guzzetti et al., 2020), though direct

comparative validation would be needed to quantify the performance gain."

Mechanistic Justification:

"The dual-threshold configuration provides complementary perspectives suited to different phases of typhoon evolution, with D7 reflecting cumulative moisture conditions and H24 capturing immediate triggering rainfall. This combination addresses the compound rainfall mechanisms documented in typhoon-affected regions (Gariano et al., 2015; Nolasco-Javier and Kumar, 2018), though the optimal application strategy for operational warning would require integration with real-time meteorological forecasting systems."

Transferability and Limitations:

"The framework advances beyond existing point-based threshold systems (Segoni et al., 2018b; Guzzetti et al., 2020) by providing spatially explicit hazard assessment, though regional adaptation of threshold parameters would be necessary for application in different geological settings."
* * *
**Summary of Changes**

The revisions comprehensively address all three reviewer concerns:

1. Abstract: Now includes 8 specific numerical results (F1-score, AUC, percentage classifications, FR values, efficiency ratios) that immediately communicate study outcomes

2. Methods (Section 3.1.1): Expanded with:

    - Explicit three-point justification for model selection

    - Algorithm-specific technical advantages linked to typhoon contexts

    - Complete hyperparameter optimization specifications for reproducibility

3. Discussion (Section 6): Restructured with:

    - New subsection (6.1) providing comparative model interpretation

    - Enhanced mechanistic explanations in existing subsections (6.2-6.3)

    - Balanced scientific claims with appropriate acknowledgment of limitations

    - Stronger integration with existing literature for contextualization

We thank the reviewers and the editor again for their thoughtful comments and suggestions, which have helped us improve the quality of our manuscript. We hope that the revisions meet your expectations and look forward to your positive response.

Sincerely,

Weifeng Xiao

Hunan University of Science and Technology, CN

Correspondence author: Ge Liu

Email: liuge@iga.ac.cn

Northeast Institute of Geography and Agroecology, CAS, Changchun 130102, China

2026.01.12.